
# GNSS radio occultation excess phase processing for climate applications including uncertainty estimation

Josef Innerkofler[1,2], Gottfried Kirchengast[1,2,3], Marc Schwärz[1,3], Christian Marquardt[4], and Yago Andres[4]

[1]Wegener Center for Climate and Global Change (WEGC), University of Graz, Graz, Austria
[2]FWF-DK Climate Change, University of Graz, Graz, Austria
[3]Institute of Physics, University of Graz, Graz, Austria
[4]European Organisation for the Exploitation of Meteorological Satellites (EUMETSAT), Darmstadt, Germany

**Correspondence:** Josef Innerkofler (josef.innerkofler@uni-graz.at)

**Abstract.** Earth observation from space provides a highly valuable basis for atmospheric and climate science, in particular also through climate benchmark data from suitable remote sensing techniques. Measurements by Global Navigation Satellite System (GNSS) radio occultation (RO) qualify to produce such benchmark data records as they globally provide accurate and long-term stable datasets for essential climate variables (ECVs) such as temperature. This requires a rigorous processing

from the raw RO measurements to ECVs, with narrow uncertainties. In order to fully exploit this potential, Wegener Center's Reference Occultation Processing System (rOPS) Level 1a (L1a) processing subsystem includes uncertainty estimation in both precise orbit determination (POD) and excess phase profile derivation.

    Here we introduce the new rOPS L1a excess phase processing, the first step in the RO profiles retrieval down to atmospheric profiles, which extracts the atmospheric excess phase from raw SI-traceable RO measurements. This excess phase processing,

for itself algorithmically concise, includes integrated quality control and uncertainty estimation, which requires a complex framework of various subsystems that we first introduce before describing the implementation of the core algorithms. The quality control and uncertainty estimation, computed per RO event, are supported by reliable forward-modeled excess phase profiles based on the POD orbit arcs and collocated short-range forecast profiles of the European Reanalysis ERA5. The quality control removes or alternatively flags excess phase profiles of insufficient or degraded quality. The uncertainty estimation

accounts both for relevant random and systematic uncertainty components and the resulting (total) uncertainty profiles serve as starting point for the subsequent uncertainty propagation through the retrieval processing chain down to the atmospheric ECV profiles.

    We also evaluated the quality and reliability of the resulting excess phase profiles based on Metop-A/B/C RO datasets for three 3-month periods in 2008, 2013, and 2020 by way of a sensitivity analysis for three representative atmospheric layers

(tropo-, strato-, mesosphere), investigating consistency with ERA5-derived profiles, influences of different orbit and clock inputs and consistency across the different Metop satellites. These consistencies range from centimeter to submillimeter levels, indicating that the new processing can provide highly accurate and robust excess phase profiles. Furthermore, cross-validation and inter-comparison with excess phase data from the established data providers EUMETSAT and UCAR revealed subtle discrepancies but overall very close agreement, with larger differences against UCAR in the boundary layer. The new rOPS



L1a processing can hence be considered capable to produce reliable long-term data records including uncertainty estimation for the benefit of climate applications.

## 1   Introduction

Satellite-based remote-sensing observations of the atmosphere, throughout the troposphere and stratosphere, constitute an
important backbone for contemporary atmospheric and climate science. With global warming on-going and its worldwide environmental and socioeconomic implications, improvement of the observational foundation of the Earth's climate system has become more important than ever (IPCC, 2021). The Global Climate Observing System (GCOS) therefore aims for the establishment and preservation of climate benchmark data records for the detection, projection, and attribution of changes in the climate system as essential (GCOS, 2021).

Observing Essential Climate Variables (ECVs) (National Research Council, 2007; GCOS, 2021; Bojinski et al., 2014), the Global Navigation Satellite System (GNSS) Radio Occultation (RO) measurements (e.g., Kursinski et al., 1996; Syndergaard, 1999; Hajj et al., 2002) qualify to provide benchmark data records, as they provide accurate and precise monitoring of ECVs, such as temperature and pressure over the troposphere and stratosphere, and tropospheric water vapor, with global coverage, long-term stability, and virtually all-weather capability (Anthes, 2011; Steiner et al., 2011, 2020).

During an occultation measurement, signals emitted by a GNSS satellite scan the atmosphere in limb sounding geometry and arrive with a time delay at the receiving RO satellite in low Earth orbit (LEO), which occurs due to the signal's refraction in the Earth's atmosphere. A vertically setting or rising occultation event is observed depending on whether the GNSS transmitter satellite sets or rises behind the Earth's horizon, from the viewpoint of the rapidly moving LEO receiver satellite.

The raw phase change measurements, obtained by a GNSS receiver aboard the LEO satellite, fundamentally can be traced
to the SI seconds, a requirement for RO measurements to serve as Fundamental Climate Data Record (FCDR) (GCOS, 2021). This is ensured by atomic clocks aboard the GNSS satellites, which are linked to a ground-based atomic clock network for monitoring and correction of the space-based clocks. In order to maintain traceability of the less stable oscillators of the LEO satellite, the clock bias is estimated along with position and velocity of the satellite within the precise orbit determination (POD) process (Montenbruck et al., 2008; Innerkofler et al., 2020). This allows accurate georeferencing of the measurements in order
to isolate the phase delay induced by the atmosphere. This so-called *excess phase* serves as key FCDR variable available from the RO measurements. Within the RO retrieval, this FCDR can be processed further to vertical atmospheric profiles of bending angle and refractivity, and subsequently to ECVs, namely temperature, pressure, and tropospheric water vapor.

The assessment of the quality and uncertainty of RO data, caused by measurement and retrieval errors, is also affected by the use of external data that facilitate the retrieval (Wee and Kuo, 2015). The inclusion of auxiliary information in the retrieval
and its influence advance with each step of the retrieval, and hence partially degrades the fundamental SI traceability of the





measurements. Evaluation at basic (low-level) RO data, in particular the excess phase data record, benefits from very small influence of external data in its determination, and therefore offers the possibility of obtaining highly accurate FCDRs.

For this reason the RO processing at WEGC underwent a substantial revision with the decisive change to start processing from raw measurement data and to independently perform POD of the LEO receiver satellites (Innerkofler et al., 2020). Prior to those changes, OPSv5.6 (Angerer et al., 2017), the former WEGC processing system, started with the supply of excess phase data along with already interpolated orbit and clock data from the COSMIC Data Analysis and Archive Center (CDAAC). The recent Reference Occultation Processing System (rOPS) (Kirchengast et al., 2016, 2018) now aims to process raw RO phase measurements into ECVs in a way which is SI-traceable to the universal time standard and which includes rigorous uncertainty propagation. This climate-quality rOPS comprises the *occultation data processing* (ODP) from raw RO measurement data (level 0) to excess phase (level 1a), atmospheric bending angles (level 1b), refractivity/dry-air profiles (level 2a), and finally to thermodynamic ECV profiles, as outlined in Figure 1 (red box). As a basis for supporting the ODP, the *daily system modeling* (DSM; green box) for observation geometry and atmospheric background modeling, and the *event system modeling* (ESM; orange box) for RO event geometry and environment modeling, as well as the simulation of RO profiles, were added.

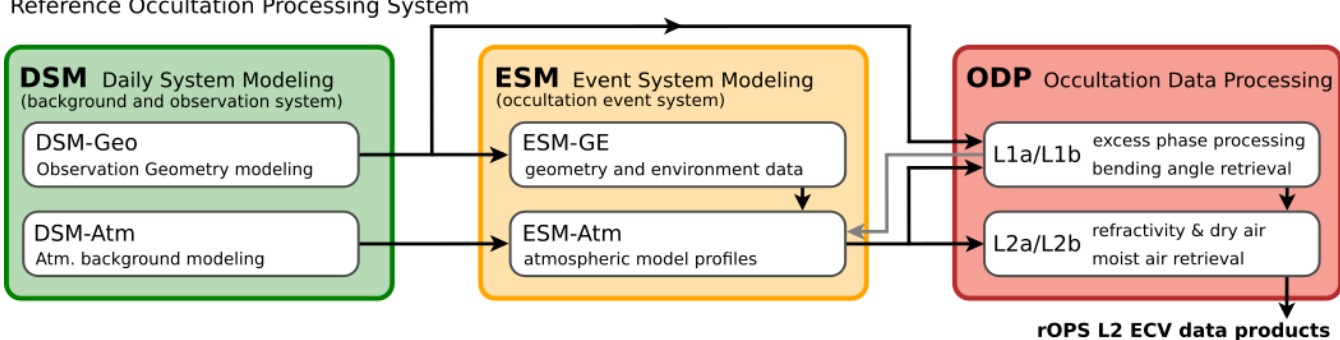

**Figure 1.** General schematic overview of parts of WEGC's rOPS relevant to this study, comprising the daily system modeling (DSM; green), the event system modeling (ESM; orange), and the occultation data processing (ODP; red). The latter outlines the main level 1 and level 2 retrieval steps (L1a to L1b) from excess phase to ECVs.

Within this study the new rOPS implementation of level 1a excess phase processing within rOPS is introduced, closing the gap between the calculation of precise orbit positions, velocities, and clock estimates (Innerkofler et al., 2020) to the retrieval of bending angles (Schwarz et al., 2018) and, subsequently, of refractivity and atmospheric profiles (Schwarz et al., 2017; Schwarz, 2018; Li et al., 2019). This includes the uncertainty estimation at excess phase level provided by the rOPS for each individual RO event.

Quality assessment of the algorithms is carried out based on RO data from the EUMETSAT Polar System (EPS) Meteorological Operational (Metop) satellite series (Luntama et al., 2008). Metop satellites provide a stable data record of RO measurements over more than a decade, which is routinely processed by two renowned processing centers: EUMETSAT as the operator of the mission (von Engeln et al., 2009) and CDAAC as independent party (Schreiner et al., 2011). Simulated





profiles extracted from ECMWF ERA5 data (Hersbach et al., 2020) support this evaluation, which is subdivided in sensitivity, statistical, and uncertainty analysis (Section 4).

The Metop satellite series consists of three flight models, Metop-A/B/C, which were launched into orbit sequentially in time (Metop-A: 2006-10-19, Metop-B: 2012-09-17, Metop-C: 2018-11-07). All three satellites are equipped onboard with the GNSS Receiver for Atmospheric Sounding (GRAS) RO instrument, developed by Saab Ericsson Space (SAAB, 2004; Klaes et al., 2007; Loiselet et al., 2000). Metop-A has then been decommissioned by November 2021 (https://www.eumetsat.int/plans-metop-end-life), while Metop-B/C are still in orbit and operational. The GRAS instrument provides dual-frequency nav-

igation and occultation tracking at 12×3 channels for L1 C/A and L1/L2 P(Y) code. Besides a zenith-looking antenna providing navigation tracking data for POD, two high-gain antennas looking in flight velocity and anti-velocity direction share 4 out of the 12 channels for recording setting and rising occultations, respectively. GRAS supports closed-loop (50 Hz) and open-loop (1 kHz) measurements (Bonnedal et al., 2010). RO data from EPS are limited to GPS observations, while second generation EPS-SG will be capable of tracking GPS, Galileo, and BeiDou signals (https://www.eumetsat.int/eps-sg-radio-occultation).

Hence, the scope of the new algorithm's evaluation in this study focuses on GPS RO data.

Following this introduction, in Section 2, the processing setup is described, the input data and their sources are summarized (Section 2.1), and the POD and background data modeling are introduced (Section 2.2). Subsequently, the excess phase processing and algorithmic description are provided (Section 3.1), complemented by description of the quality control and estimation of measurement uncertainties (Section 3.2). The results are presented and discussed in Section 4, before ending

with a conclusion section (Section 5).

## 2    Processing setup & system modeling

Processing raw RO measurements to atmospheric excess phase profiles, termed also level 1a (L1a) processing, follows a basic algorithmic sequence sometimes also referred to as "calibration" of the measurements (Hajj et al., 2002). In practice, inclusion of rigorous quality control, uncertainty estimation, and proper data preparation for the subsequent L1b bending angle retrieval

algorithm make this L1a processing a fairly complex task. Within rOPS, the excess phase processing is divided into two parts: (1) the derivation of the "raw" excess phase (the main "calibration" algorithm) and (2) the quality control and uncertainty estimation of the derived excess phase profiles. Along with these two subprocesses, a series of steps, involving ESM-GE and ESM-Atm (cf. Figure 1), complement the excess phase computation.

Figure 2 summarizes the relations and workflow between those rOPS subsystems most relevant to the L1a excess phase

processing. In a first step, external input data have to be acquired and prepared to satisfy the rOPS data interface. The DSM supplies precise and accurate orbit and clock data of LEO receiver and GNSS transmitter satellites, including estimates of systematic and random uncertainty. Subsequently, the ESM-GE models reference locations of all possible RO events and provides additional information on the geometry and environment, while the "raw" excess phase profile processing delivers vertical profiles of measured RO data. With the event information and the raw L1a data, atmospheric excess phase model

profiles can be provided by the ESM-Atm. These model profiles are then used for the L1a quality control and uncertainty





estimation, yielding "final" quality-controlled excess phase profiles, which serve as input for the ODP L1b retrieval processing. In the following, we take a closer look at each of these steps relevant to the L1a processing after summarizing the relevant input data.

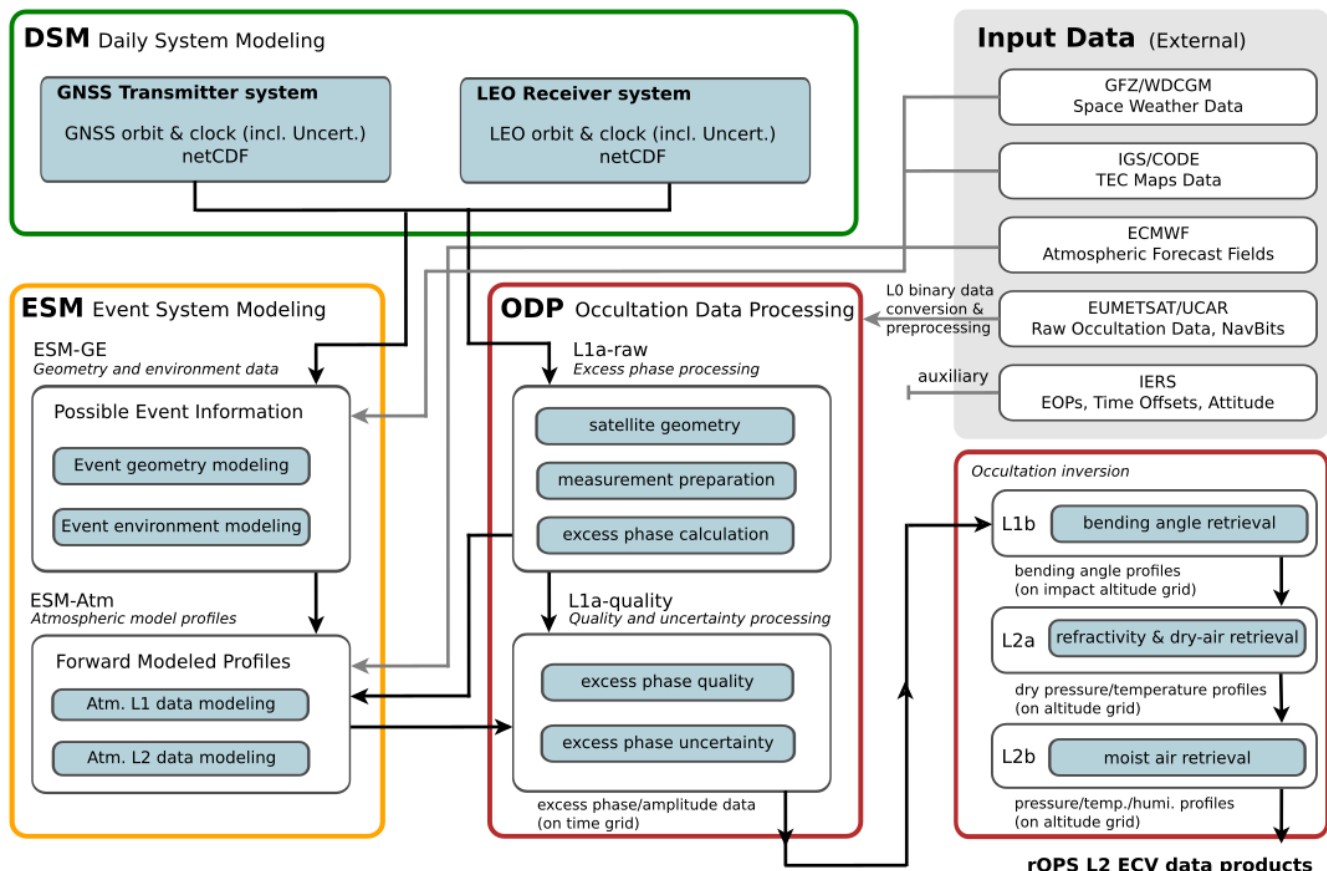

**Figure 2.** Schematic overview of parts of WEGC's rOPS relevant to this study, comprising the daily system modeling (DSM; green), the event system modeling (ESM; orange), and the occultation data processing (ODP; red). Main details of subsystems are identified, with focus on the ODP L1a processor, together with the major input data sources and output flow towards the L1b and L2 (L2a, L2b) retrieval chain from excess phase to ECVs.

## 2.1 Input data preparation

The RO data processing in general, and more specifically to this study the WEGC ODP-L1a excess phase data processing, relies on various input data sources. Besides the main observational data measured by the satellite's occultation antenna(e), also space weather data, atmospheric reanalysis and forecast data, satellite antenna specifications, GNSS navigation bit data, and auxiliary data, comprising Earth orientation data and time offsets, are necessary input data to RO excess phase processing





and its supporting ESM system. Table 1 provides a concise overview of these external data sources used for the L1a processing
within rOPS and is followed by a description of the application of these data.

**Table 1.** External input data used by the different subsystems of the rOPS ODP-L1a processing system. Data is provided by Crustal Dynamics
Data Information System (CDDIS), German Research Centre for Geosciences (GFZ), Geomagnetic Laboratory (GEOLAB) of Natural
Resources Canada, ECMWF, International GNSS Service (IGS), EUMETSAT, and International Earth Rotation and Reference Systems
Service (IERS). For details on the specific data types see text.

| data type | provider | resources |
|---|---|---|
| *ESM-GE* | | |
| TEC maps | IGS/CDDIS | https://cddis.nasa.gov/archive/gnss/products/ionex; Hernández-Pajares et al. (2009) |
| Kp index | GFZ | ftp://ftp.gfz-potsdam.de/pub/home/obs/kp-ap; Matzka et al. (2021) |
| Solar flux F10.7 | GEOLAB | ftp://ftp.geolab.nrcan.gc.ca/spaceweather/solar_flux/daily_flux_values; Tapping (2013) |
| *DSM-Atm* | | |
| Atmospheric Fields | ECMWF | Climate Data Store Application Program Interface (CDS); ECMWF (2019) |
| *OPD-L1a (raw processing)* | | |
| GNSS ANTEX | IGS | https://files.igs.org/pub/station/general/igs14.atx; Rebischung and Schmid (2016) |
| LEO ANTEX | N/A | from satellite providers; format: https://files.igs.org/pub/station/general/antex14.txt |
| Occultation data | EUMETSAT | https://eoportal.eumetsat.int, EUMETSAT (2016) |
| Navigation bit data | EUMETSAT | https://eoportal.eumetsat.int, EUMETSAT (2018) |
| *auxiliary* | | |
| EOP, time offsets, ... | IERS | https://datacenter.iers.org/eop.php |

In this study we used three multi-months time periods as basis for the assessement, each comprising 3 months: July to
September 2008 (Metop-A); July to September 2013 (Metop-A/B); December 2019 to February 2020 (Metop-A/B/C). These
are representative for different solar cycle and summer/winter conditions over more than a decade from 2008 to 2020. Data
availability of the respective study periods is limited by the different launch dates of the satellites A, B, and C (cf. Section 1).

Compared to other RO receivers, the GRAS receiver onboard the Metop satellites delivers raw-level measurement data
in form of chunks of numerically controlled oscillator (NCO) time-tagged phase as well as in-phase and quadrature (I/Q)
components, instead of already connected phase and amplitude as function of time (SAAB, 2004; Schreiner et al., 2011).
Since the computation of these variables is time intense, and some data distribution limitations for original level 0 data by
EUMETSAT do apply, WEGC decided to reconstruct the raw variables from the EUMETSAT L1a data (EUMETSAT, 2016).

This includes reconstruction of the raw measurement time stamps and un-differencing of the NCO phase. It was validated
against processing based on EUMETSAT level 0 data, that such correctly performed raw-level measurement reconstruction
does not show any non-negligible differences to the original data. We note that CDAAC developed their own phase and



amplitude calculation for GRAS data, which as well shows good agreement with data generated by EUMETSAT (Schreiner et al., 2011).

For evaluation and validation of the calculated atmospheric excess phase profiles, and *inter alia* also of the proper implementation of the rOPS ODP-L1a processing, a careful intercomparison with independently calculated profiles is indispensable. In this study publicly available excess phase profiles from EUMETSAT (https://eoportal.eumetsat.int) and CDAAC (https://cdaac-www.cosmic.ucar.edu) have been used for such intercomparison (Sect. 4.2). Additionally, forward-modeled profiles computed from ERA5 reanalysis data served as reference for a sensitivity analysis for the excess phase profiles calculated

with the rOPS (Sect. 4.1.1).

Reanalyses data are widely used in atmospheric sciences applications and also serve as an integral component in WEGC's rOPS system. The reference dataset used in this study is the ERA5 reanalysis dataset, computed with the integrated forecasting system IFS of the European Center for Medium-Range Weather Forecasts (ECMWF) (ECMWF, 2016, 2019; Hersbach et al., 2019, 2020). Within rOPS, ERA5 short-range *forecast* data are used for the provision of model and background profiles in

order to facilitate the ODP retrieval and to derive quality measures and support estimation of the uncertainty of the observed profiles at each step of the retrieval, while the ERA5 *analysis* data are used for the provision of reference profiles for validation of the RO retrieval.

The employed short-range forecast is essentially a pure model state carrying no direct information from assimilated observational data. This is an important aspect in order to keep the final rOPS ECVs uncorrelated with the evaluation (analysis)

dataset, given that RO data are assimilated into ECMWF's (re)analyses as of late 2006 (Healy, 2007; Hersbach et al., 2020). With a horizontal resolution of about 30 km, and 137 vertical levels from the surface up to about 80 km, the ERA5 reanalysis data suite the RO measurement characteristics well. The interpolation strategy to produce collocated model, background, and reference profiles is summarized in Section 2.2.3.

For the ESM-GE processing, ionosphere and space weather information are incorporated in order to complement the en-

vironment information for each RO event. These data comprise total electron content (TEC) maps, Kp indices describing disturbances of the Earth's magnetic field, and solar flux data measuring the radio emission from the Sun at a wavelength of 10.7 cm as a proxy for the solar activity. These can support quality evaluation and, in particular, serve as input for higher-order ionospheric correction in L1b processing (Danzer et al., 2020, 2021; Liu et al., 2020). In addition, GNSS and LEO antenna offset information is used to accurately model the event locations in ESM-GE and to correct the location of the observations in

OPD-L1a. For the GNSS, these data are provided by the International GNSS Service (IGS) using the antenna exchange format (ANTEX; https://kb.igs.org/hc/en-us/articles/216104678-ANTEX-format-description).

For the LEO receiver satellites, different sources need to be exploited depending on the RO mission and satellite data provider. At WEGC all relevant LEO antenna information is merged in a custom receiver antenna file using as well the ANTEX format. GNSS navigation bit data for measurement signal demodulation are obtained from EUMETSAT (EUMETSAT, 2018).

Variable Earth geodetic data, such as Earth orientation parameters (EOP), leap seconds, and time offsets, serve as general input in support of coordinate and time transformations performed as part of the rOPS L1a processing.





## 2.2 Daily and event-based system modeling

The system modeling part of the rOPS, in preparation for the ODP core system, consists of two main components: (1) DSM: the daily system modeling of the *observation system geometry* and preparation of *atmospheric background* fields and (2) ESM:

the occultation event modeling of *geometry and environment* modeling and derivation of corresponding *atmospheric model profiles* (Figure 2).

### 2.2.1 Observation geometry modeling

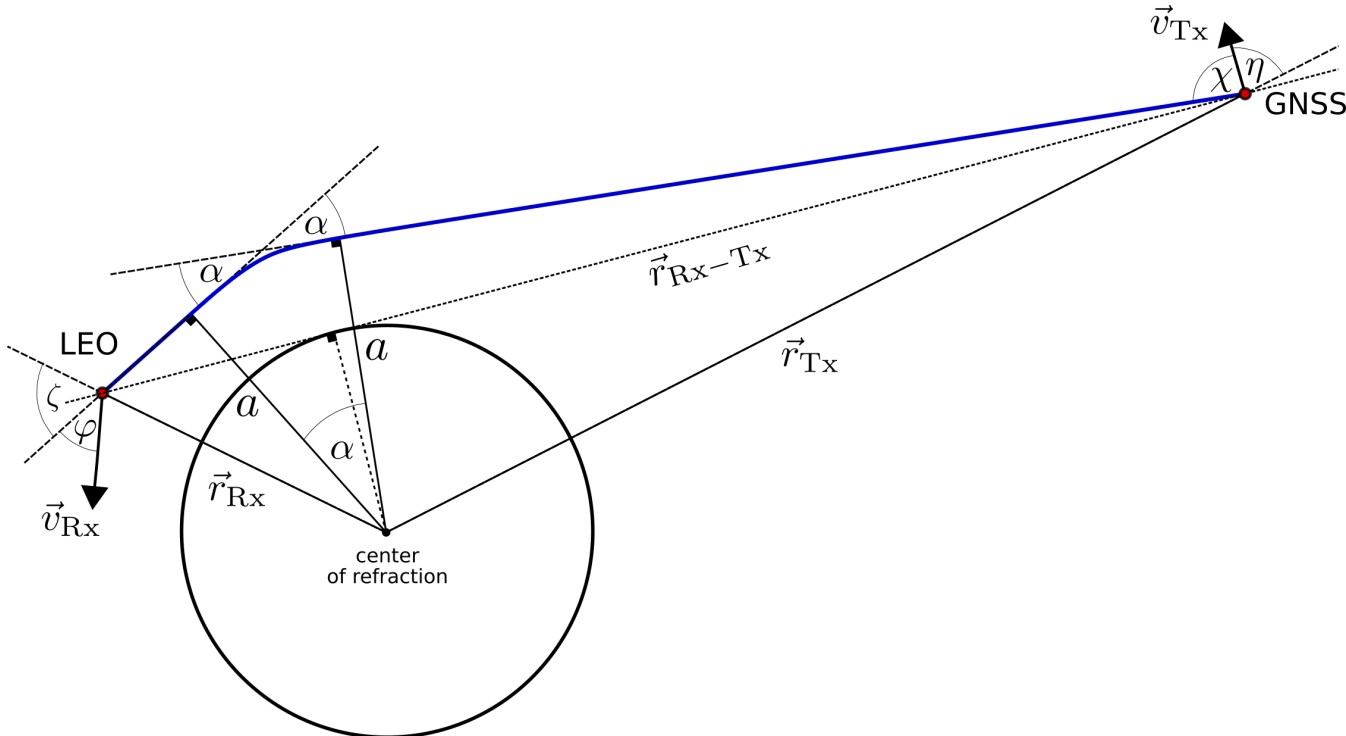

**Figure 3.** Occultation event geometry, defining important location and angular variables of an RO event. Adapted from Pirscher (2010).

Accurate and precise knowledge of the location and time of GNSS signal transmission and reception is fundamental to RO processing and the subsequent derivation of highly accurate and long-term stable ECVs. For this reason, a novel multi-system

setup for POD was introduced by Innerkofler et al. (2020) that builts on independent orbit solutions from three different processing runs for each day. Embedded in the DSM observation geometry modeling of rOPS, the system also provides attributed measures for quality and estimated uncertainties, in order to enable long-term stable and highly consistent LEO orbit processing and analysis.

Within the DSM-Geo, the LEO POD of RO receiver satellites is performed in parallel employing the two independent

software packages of Bernese v5.2 (Dach et al., 2015) and NAPEOS v3.3.1 (Springer, 2009). The calculations are based





on GNSS pseudocode and carrier phase measurements obtained by the RO satellite's zenith antenna and GNSS orbit and clock data products from the Center for Orbit Determination in Europe (CODE). One additional orbit solution is derived with transmitter orbit and clock data from the International GNSS service (IGS) using Bernese. This POD setup enables mutual consistency checks of the calculated orbit solutions and is used for monitoring the quality of the "primary" orbit solution (the

one derived with Bernese-CODE used for further RO processing) as well as for position and velocity uncertainty estimation, including estimated systematic and random uncertainties. For more details on this rOPS POD processing the reader is referred to Innerkofler et al. (2020).

The additional uncertainty output produced by DSM-Geo exceeds the limitation of the well-established format definitions used for the exchange of orbit data, e.g. SP3 (https://files.igs.org/pub/data/format/sp3c.txt). We therefore store the final output

data of the POD in a specially designed netCDF file that contains orbit and clock data for each daily 24 hour orbit arc per satellite. In total this results in nominally 32 GPS transmitter and 1 RO receiver satellite file per day. The content comprises the center-of-mass (COM) satellite positions and velocities in an Earth-centered, Earth-fixed (ECEF) coordinate system as well as satellite clock biases. Adequate for the intended purpose of RO processing (confirmed by comprehensive sensitivity checks), the GNSS orbit data are stored with a sampling rate of 15 minutes, whereas the clock-bias data and LEO orbit data are stored

at a sampling rate of 30 seconds. All these daily orbit-arc data are stored together with their estimated random and systematic uncertainty.

Table 2 provides an overview of the core variables supplied by the DSM-Geo POD processing, which subsequently serve as input for the OPD-L1a processing. For the excess phase processing these data need to be aligned in time (interpolation of RO measurement time stamps and correction of clock biases) and location (application of RO antenna offsets). The preparation

and necessary steps for this processing are discussed further in Section 3.1.

**Table 2.** Orbit and clock data output from the rOPS POD processing, including the associated sampling rates and corresponding estimates of random ($u_r$) and systematic uncertainty ($u_s$).

|  | Variable | Unit | Description |
|---|---|---|---|
| **LEO** | $r^r$ | m | COM position of LEO satellite, each 30s including estimates of $r^r_{u_r}$, $r^r_{u_s}$ |
|  | $v^r$ | m s$^{-1}$ | COM velocity of LEO satellite, each 30s including estimates of $v^r_{u_r}$, $v^r_{u_s}$ |
|  | $\delta t^r$ | s | clock bias (30s) including $\delta t^r_{u_r}$ |
| **GNSS** | $r^t$ | m | COM position of LEO satellite, each 15min including estimates of $r^t_{u_r}$, $r^t_{u_s}$ |
|  | $v^t$ | m s$^{-1}$ | COM velocity of LEO satellite, each 15min including estimates of $v^t_{u_r}$, $v^t_{u_s}$ |
|  | $\delta t^t$ | m | clock bias (30s) including $\delta t^t_{u_r}$ |





### 2.2.2 Event system modeling

Provided the positions and velocities of the transmitter and receiver satellites, calculated by the DSM-Geo, a reference event location of an occultation is determined in the rOPS by geometrical constraints only, independent of any atmospheric state or RO measurement data. The selected reference location of an event is defined by the mean event time and Earth's ellipsoidal surface as the point when the straight line connection between receiver and transmitter satellite is tangent to the Earth's surface (WGS-84/EGM2008, cf. Figure 3 for measurement geometry). This uniquely-defined point in space and time, also referred to as *mean tangent point* of the RO event, is determined after the GNSS orbit is interpolated to the 30-s-sampled LEO orbit time stamps (and locally using fine-interpolation for accurately determining the point). For compact identification, each event is uniquely encoded using a 64-bit integer representation compiled from mean event time, transmitter system and transmitting-satellite identifier, as well as receiver identifier.

The event geometry modeling is complemented by the event environment modeling comprising surface and atmosphere data, as well as ionosphere and space weather information extracted for each reference location. Apart from the mean tangent point, which per definition is at a straight-line tangent height (SLTH) of 0 km, the same information can be additionally retrieved for any other non-zero SLTH. In rOPS this is done additionally at an SLTH of 90 km, 80 km, 70 km, 60 km, and −250 km, respectively. Extracted parameters and variables at these auxiliary locations visited during the RO event is valuable for contextual grouping of events, event distribution analyses, ionosphere/space weather senstivity analyses, and is as well needed in the subsequent ODP processing, e.g., as part of higher-ionospheric correction (Danzer et al., 2021; Liu et al., 2020).

The mean-tangent-point location derived in this way has been proven to be very adequate and corresponds to real RO event tangent-point trajectory locations near the tropopause (Foelsche et al., 2011). This offers the advantage that purely geometrical analyses, like for geographic RO event distributions, can be done independent of RO data and it enables rigorous location consistency of forward modeling (simulating RO profiles) and retrievals (analyzing observed profiles) for each and all individual events. Figure 4 illustrates the global RO event distribution for Metop-A/B on July 1, 2013. On principle the number of modeled events exceeds the number of observed events. The calculated data of the ESM-GE are provided as dedicated output file of rOPS, then further used by the subsequent ESM subsystems and the ODP-L1a processing.



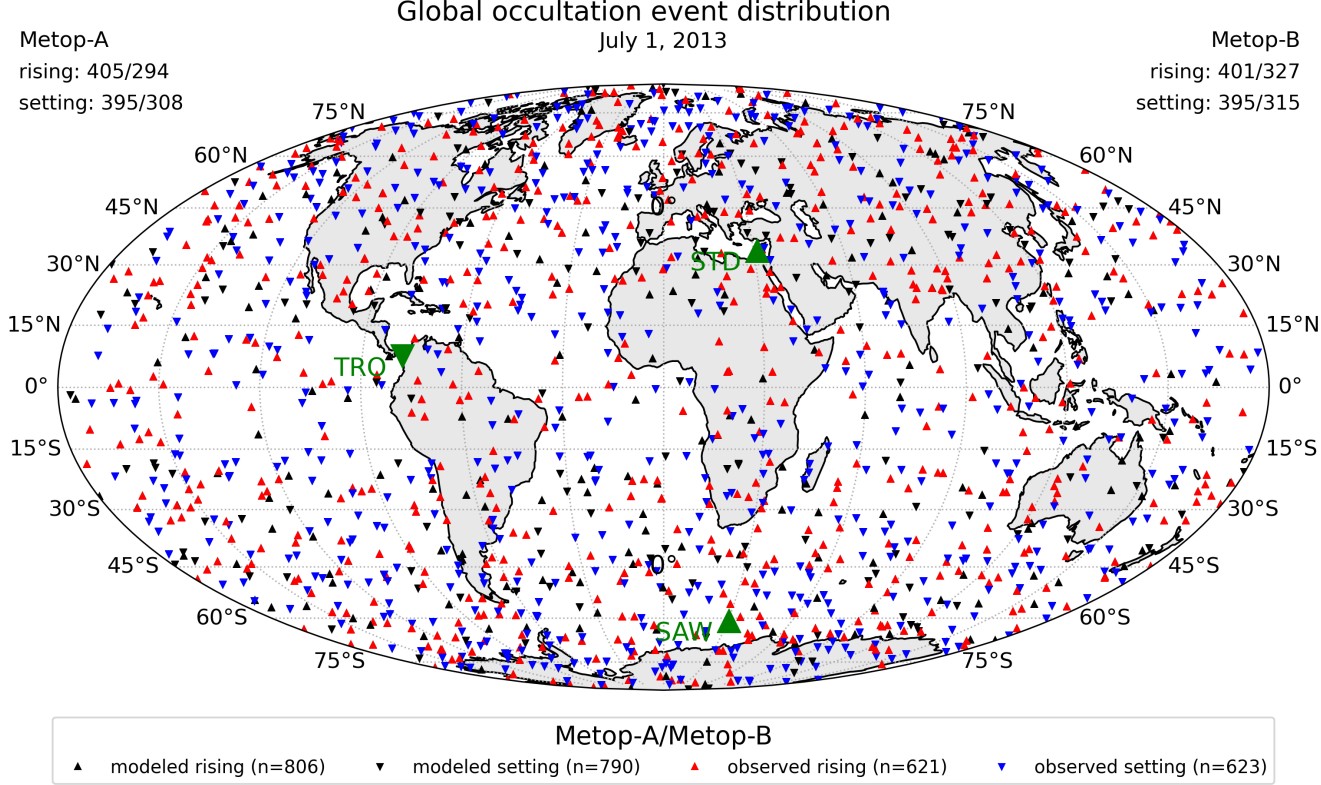

**Figure 4.** Global distribution of Metop-A/B RO events on 1 July 2013. Setting events (upside-down triangles) and rising events (upright triangles) are separately identified and the numbers of modeled/processed events noted as part of the legend information. Three example events used for illustrating the subsequent description and results are as well separately identified (large green triangles), with more information on them summarized in Table 3).

### 2.2.3 Atmospheric model profiles

With knowledge of the simulated but realistic event location of the measurements from the ESM-GE, corresponding atmospheric profiles can be forward-modeled upward along the ODP processing chain from level 2 to level 1. Figure 5 illustrates the forward-modeled results, extracted from ERA5 data at the respective locations, for the three example events (see Figure 4 and Table 3), for refractivity ($N$), bending angle ($\alpha$), and excess phase ($L$), respectively, as well as the difference profiles between the forward-modeled and related observed excess phase data. These example profiles serve to represent standard (STD), tropical (TRO), and sub-arctic winter (SAW) conditions. Given the simulated event locations, it is ensured that for each observed RO event, a corresponding consistent set of forward-modeled profiles is made available at each step from refractivity to excess phase.

The ERA5 reanalysis fields, introduced in Section 2.1 above, globally provide temperature, pressure, and water vapor information at any desired location and time of interest. Hence, starting from ERA5 fields, atmospheric profiles of these variables



**Table 3.** Selected Metop-A RO example events, representing standard (STD), tropical (TRO), and sub-arctic winter (SAW) conditions.

| No. | Label | GPS-ID | Mission | Latitude | Longitude | Time | Characterization |
|-----|-------|--------|---------|----------|-----------|------|------------------|
| 1 | STD | 03 | Metop-A | 33.54 N | 31.24 E | 17:36:19 | standard atmosphere |
| 2 | TRO | 20 | Metop-A | 7.45 N | 77.96 W | 13:22:12 | tropical atmosphere |
| 3 | SAW | 14 | Metop-A | 60.84 S | 30.61 E | 06:42:07 | southern atm. winter |

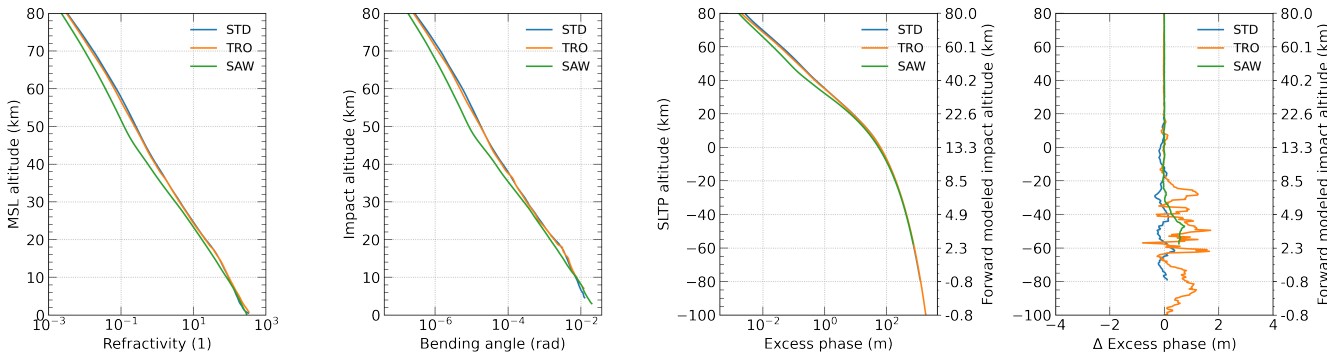

**Figure 5.** Forward-modeled example profiles from refractivity (left) to bending angle (middle-left) and excess phase (middle-right), respectively, and difference between modeled and observed excess phase (right). Results for all three example events defined in Table 3 are shown.

are extracted, by interpolation in time and space to the RO event location (mean-tangent point location, extracting vertical profiles at these locations). Since the ERA5 fields are used in a 6-hourly-resolved form, the closest time layer happens to be always within 3-hours of the occurrence of the mean RO event time; a time difference which is sufficient to model the semi-diurnal cycle (Scherllin-Pirscher et al., 2011b,a). Horizontal interpolation is performed by using a cubic-polynomial in-
terpolation technique. For vertical interpolation to a fixed-altitude grid $z$, natural cubic-spline interpolation is used $T$ and $ln(p)$, while $q$ is interpolated linearly.

Refractivity profiles are then derived, based on the interpolated variables, employing the ionosphere-free first order relationship (Smith and Weintraub, 1953; Kursinski et al., 1997). Given the refractivity profile as function of the fixed vertical grid $N(z)$ level 1 profiles of bending angle $\alpha$, Doppler shift $D$, and excess phase $L$ can be calculated. In a first step the
impact altitude grid $z_a(z) = z + 10^{-6}N(z) \cdot (z + U_G + R_C)$ and the impact parameter grid $a_z(z_a) = z_a(z) + U_G + R_C$ are computed consistent with the altitude grid $z$, with the geoid undulation $U_G$ and radius of curvature $R_C$, both at the occultation event location. Additionally the log-refractive index profile $n^{ln}(a_z) = ln(1 + 10^{-6}N(z))$ and its gradient profile $n_{da}^{ln}(a_z)$ are calculated.

Using the refractive index gradient profile the bending angle profile $\alpha(a_z)$ is then derived using an inverse Abelian integral
transformation (Fjeldbo and Eshleman, 1965; Fjeldbo et al., 1971) solving for all grid levels $a_i$ from $a(z_{top})$ to $a(z_{bot})$. The actual calculation is performed using the implementation in rOPS (Syndergaard and Kirchengast, 2016) in combination with





the baseband method (cf. Kirchengast et al. (2018)). Subsequently, the bending angle profile is mapped to a strict 50 Hz measurement grid including a vertical mapping from impact altitude $\alpha(a)$ to time $\alpha(t)$. This mapping requires the knowledge of orbit position of transmitter ($r_{\mathrm{Tx}}$) and receiver ($r_{\mathrm{Rx}}$) satellites at measurement time and demands an iterative numerical 255 solution (Proschek et al., 2011).

The mapped bending angle $\alpha(t)$ is then used to calculate the Doppler shift which can in turn be further integrated to excess phase. Based on the occultation geometry (cf. Figure 3) and the corresponding angles $\chi$, $\eta$, $\varphi$, $\zeta$, as well as the position and velocity of receiver ($r_{\mathrm{Rx}}$, $v_{\mathrm{Rx}}$) and transmitter ($r_{\mathrm{Tx}}$, $v_{\mathrm{Tx}}$) satellites and their distance to each other $\mathrm{d}r_{\mathrm{RxTx}}$ the Doppler shift can be calculated as follows:

$$D(t_i) = [v_{\mathrm{Rx}}\cos\varphi(a_i) - v_{\mathrm{Tx}}\cos\chi(a_i)] - \frac{\mathrm{d}r_{\mathrm{RxTx}}(t_i)}{\mathrm{d}t}, \text{ with} \tag{1}$$

$$\chi(a_i) = (\pi - \eta) - \arcsin\left(\frac{a_i}{r_{\mathrm{Tx}}}\right) \quad \text{and} \quad \varphi(a_i) = \zeta - \arcsin\left(\frac{a_i}{r_{\mathrm{Rx}}}\right). \tag{2}$$

The modeled Doppler shift $D(t_i)$ is limited to the proportion of the entire Doppler shift which is induced by the atmosphere, while those parts which come from the movement of the satellites relative to each other are not considered.

Finally, the modeled excess phase $L(t_i)$ can be calculated as function of time by using the relation with the previously calculated Doppler shift $D(t_i)$:

$$L(t_i) = \int_{t_{bot}}^{t_{top}} D(t_i)\,\mathrm{d}t + L(t_{top}). \tag{3}$$

To integrate the excess phase change (or Doppler shift), in order to obtain the excess phase path, we apply the Simpson's rule for the numerical integration of Equation 3. The integration starts at the top of the measurement, where $D$ is very small 270 and errors in the initialization are almost vanishing relative to contributions at lower atmospheric levels. The scale height $H$ times the bending angle $\alpha$ at the uppermost level serves as integration constant $L(t_{top}) = H_{top} \cdot \alpha_{top}$, which is considered an adequate approximation for the integration of the excess phase (Rieder and Kirchengast, 2001; Kirchengast et al., 2018).

## 3 Excess phase processing and algorithmic description

### 3.1 Excess phase processing

Numerous studies have described the RO retrieval chain in detail (e.g., Kursinski et al., 1997; Rieder and Kirchengast, 2001; Hajj et al., 2002; Kuo et al., 2004) and have shown the high accuracy of RO data, particularly in the upper troposphere and lower stratosphere region (e.g., Rocken et al., 1997; Gobiet et al., 2007; Ho et al., 2012; Steiner et al., 2020). More specifically, also RO level 1a processing from raw occultation measurements and precise orbit and clock data of transmitter and receiver satellites to excess phase level has been discussed in detail (e.g., Hajj et al., 2002; Beyerle et al., 2005; Schreiner et al., 2010; 280 Bai et al., 2017), since accurate and precise excess phase data are the indispensable basis for SI-traceability and co-determine the quality of derived ECVs. Here the implementation of WEGC's new rOPS-ODP-L1a excess phase processing is described, preceded by a short discussion of GNSS signal structure and signal tracking modes.



### 3.1.1 GNSS observables

In principle, the concept of GNSS measurements relies on measurements of ranges between transmitter and receiver which
are derived from measured time (code pseudoranges) or phase differences (phase pseudoranges) relative to the transmitted
electromagnetic signal. Since the measured ranges are affected by transmitter and receiver satellite clock errors, they are
labeled pseudoranges. Also, more commonly, simply referred to as code and (carrier) phase measurements, those observables
constitute basic measurements made by GNSS receivers aboard RO satellites, where the latter (carrier phase) are more precise
and therefore favored in RO processing. This is intrinsic to the chip length of the code modulated onto the carrier phase (e.g.
GPS, coarse/acquisition (C/A) code: $\lambda \approx 293$ m; precision (P) code: $\lambda \approx 29$ m), which is accurate to 1-0.1%, while carrier
phase measurements on typical GNSS wavelength (e.g. GPS, L1: $\lambda \approx 0.19$ m; L2: $\lambda \approx 0.24$ m) allow millimetric precision
(Teunissen and Montenbruck, 2017; Hofmann-Wellenhof et al., 2008).

The enhanced precision of the carrier phase measurements comes with the disadvantage of ambiguity of the measurement,
since at the time of the initial signal acquisition the number of full carrier wavecycles ("integer cycles") between the receiver
and the transmitting satellite is not known. Although this is a problem for classical navigation and positioning applications, for
atmospheric RO, interested in the relative change of the signal for vertical atmospheric profiling, it does not pose a problem
(Teunissen and Montenbruck, 2017).

Phase measurements are generally performed using a phase-locked loop (PLL), where the receiver generates an internal
replica of the GNSS signal, aligns it with the incoming carrier phase and measures it while keeping track of the changes in
full cycles and the fractional shift (I/Q). Within PLL tracking, the internally generated signal is adjusted using a feedback
loop on basis of previous measurements, which works fine with sufficiently high SNR and weak atmospheric disturbances
in the atmospheric regions above the lower troposphere. In the lower troposphere, however, the complicated structure of RO
signals, caused by multipath propagation or even ducting in the moist lower troposphere, leads to a loss of lock or to biases
in the measured signals as well as to late signal acquisitions for rising occultations may miss the moist lower troposphere
(Sokolovskiy, 2001; Sokolovskiy et al., 2006; Ao et al., 2009).

In order to overcome these issues, RO signals are measured using a delay-locked loop (DLL) when disturbed atmospheric
conditions interfere with closed-loop (CL) tracking using a PLL. This open loop (OL) tracking mode is not supported with a
feedback loop including the measured signal (and therefore less prone to be affected by disturbed atmospheric conditions), but
model-aided based on orbit, receiver clock drift, and estimated Doppler shift and delay, deduced from a real-time navigation
solution (e.g., Ao et al., 2009). This onboard computation of the OL model, including the prediction of the code pseudorange, is
employed by the majority of GNSS-RO receivers currently in space. Note that RO measurements performed using OL tracking
need to be demodulated from GNSS navigation data in post-processing (Sokolovskiy, 2001; Sokolovskiy et al., 2006).

The GRAS receiver, in contrast, continues C/A code tracking in OL mode and only uses an OL model for carrier phase
tracking. In addition, another method to overcome the shortcomings of CL tracking in the lower troposphere is to use moderate
wideband digital recording, a technique which features a sufficient bandwidth to capture the spectrum of the excess Doppler
in the lower troposphere (Ao et al., 2009). For GRAS onboard of Metop (Bonnedal et al., 2010) this was realized by the





implementation of the raw sampling (RS) tracking mode, scanning the GNSS signal with a frequency of 1 kHz (SAAB, 2004; Gorbunov et al., 2011; Zus et al., 2011). On the one hand this bears the advantage that only geometric effects need to be modeled in the tracking loop but at the same time data storage requirements increase significantly compared to 50 Hz OL

tracking featuring narrow band digital recording) (Sokolovskiy, 2001; Ao et al., 2009).

Independent of the tracking mode employed, the RO technique aims to extract atmospheric and ionospheric delays from the total measured phase. The measured phase between GNSS transmitter and LEO receiver satellites, denoted with the superscript $t$ and the subscript $r$, respectively, can be modeled for a given GNSS signal frequency summing-up the following terms as function of signal reception time $t_r$ at the LEO satellite and transmission time $t^t$ at the GNSS satellite (Schreiner et al., 2010):

$$
\begin{aligned}
L_r^t(t_r, t^t) \quad &= \rho_r^t(t_r) + c[\delta t_r(t_r) + \delta t_{r,rel}(t_r)] - c[\delta t^t(t^t) + \delta t_{rel}^t(t^t)] + \lambda N_{r,amb}^t \\
&\quad - \delta \rho_{r,ion}^t(t_r) + \delta \rho_{r,atm}^t(t_r) + \delta t_{r,rel}^t(t_r, t^t) + \epsilon,
\end{aligned}
\tag{4}
$$

where $L_r^t(t_r)$ denote the total observed carrier phase in units of length, $\rho_r^t(t_r, t^t)$ the geometrical distance between the transmitter and receiver satellites, $\delta t_r(t_r)$ the receiver clock error (or bias/offset), $\delta t^t(t^t)$ the transmitter clock bias (or offset), $\delta t_{r,rel}(t_r)$ the special relativistic effect on the receiver clock, $\delta t_{rel}^t(t^t)$ the special relativistic effect on the transmitter clock, $\delta \rho_{r,ion}^t(t_r)$ the ionospheric delay, $\delta \rho_{r,atm}^t(t_r)$ the neutral atmosphere delay, $\delta_{r,rel}^t(t_r)$ the general relativistic gravitational de-

lay, $\lambda N_{r,amb}^t$ the integer phase ambiguity with the signal, wavelength $\lambda$, speed of light in vacuum $c$, and $\epsilon$ the phase noise and residual errors.

### 3.1.2    Atmospheric and ionospheric excess phase

For the purpose of remote sensing of the Earth's atmosphere the atmospheric influences, comprising effects of the neutral atmosphere $\delta \rho_{r,atm}^t$ and the ionosphere $\delta \rho_{r,ion}^t$, need to be separated from the phase model as specified above. Thus, all terms

in Equation (4) apart from the neutral atmosphere delay and ionospheric delay need to be modeled or removed in order to isolate the atmospheric and ionospheric excess phase:

$$
\begin{aligned}
\Delta L_r^t(t_r) \quad &= \delta \rho_{r,atm}^t(t_r) + \delta \rho_{r,ion}^t(t_r) + \epsilon \\
&= L_r^t(t_r) - \rho_r^t(t_r) - [\delta t_r(t_r) + \delta t_{r,rel}(t_r)] - c[\delta t^t(t^t) + \delta t_{rel}^t(t^t)] - \delta t_{r,rel}^t(t_r) + \lambda N_{r,amb}^t + \epsilon.
\end{aligned}
\tag{5}
$$

The ionospheric part can then later on be removed using a linear combination of the dual-frequency measurements (Schreiner et al., 2010) or more refined RO-tailored corrections at bending angle level (Vorob'ev and Krasil'nikova, 1994; Ladreiter and

Kirchengast, 1996; Liu et al., 2013; Healy and Culverwell, 2015; Liu et al., 2020). As depicted in Figure 6, the separation of the excess phase includes a series of corrections to the measurement geometry and the measured signal itself. In the following the necessary steps for the excess phase calculation and the implementation in rOPS are discussed in more detail.

### 3.1.3    Satellite geometry - orbit and clock corrections

As a first step of the OPD-L1a excess phase determination, the accurate space-time points of the signal's transmission and

reception need to be determined. Therefore, the low-rate GNSS and LEO orbit data need to be corrected and then interpolated





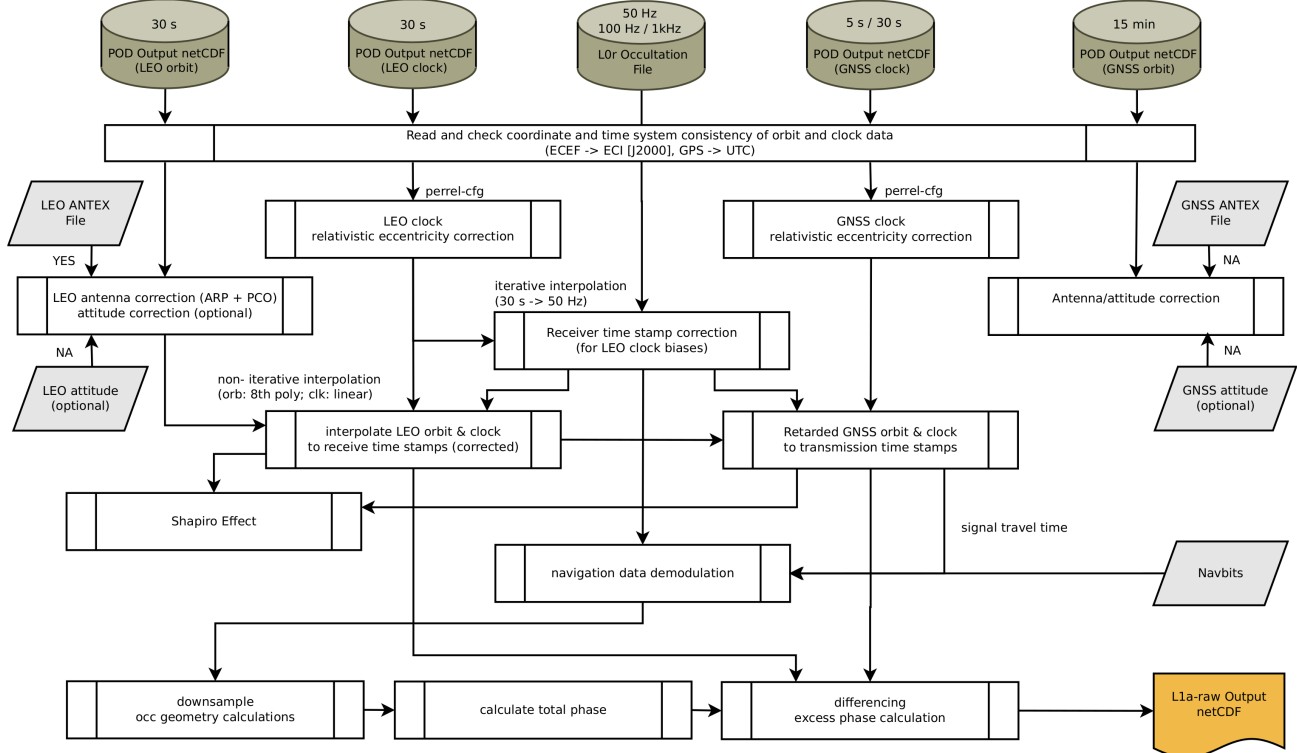

**Figure 6.** rOPS-ODP-L1a excess phase processing algorithm workflow. For description and explanations see text.

to the high-rate RO measurement time stamps. This process includes the correction for receiver and transmitter clock biases, relativistic corrections, the calculation of the signal travel time, antenna offset corrections, and the calculation of the geometric distance between the receiver and transmitter satellite based on the corrected positions and velocities of the GNSS and LEO satellites (see Figure 6).

At the beginning of the ODP-L1a processing all the necessary input data (cf. Section 2.1) are read and checked for their consistency. Starting with precise orbit and clock data from the ESM-Geo, the coordinate and time systems of the data are examined. The positions and velocities of the satellites are converted from an Earth-Centered, Earth-Fixed (ECEF) to an Earth-Centered Inertial (ECI) reference frame. That way, all calculations are independent from the Earth's rotation correction (Sagnac effect; Ashby (2004)), which has to be taken into account when working in a rotating, Earth-fixed reference frame.

More specifically, we use the J2000 coordinate system as realization of the ECI reference frame for all calculations.

The atomic clocks aboard the GNSS satellites are considered stable over the short duration of an occultation event of approximately 1 to 2 minutes with an accuracy of between $10^{-11}$ and $10^{-12}$; more up-to-date clocks even with higher accuracy (Griggs et al., 2015; Hauschild et al., 2013). In contrast to the GNSS satellites, the RO LEO missions do not carry atomic clocks on board, hence timing is in principle less accurate. However, most RO missions, including Metop-A/B/C, use ultra-stable





quartz oscillators that are likewise highly accurate over the short term of RO events. The LEO clock errors $\delta t_r$ (bias estimates along time) are estimated along with the orbit determination process at WEGC, while the GNSS clock bias estimates $\delta t^t$ are provided as part of the orbit data products by the CODE (or IGS) analysis center. The selection of a suitable differencing method, applied later in the excess phase calculation (see Section 3.1.4), is dependent on the stability of the LEO clock.

Satellites and their clocks move with considerable speed in the Earth's gravity field, and orbit at fairly high altitudes above

surface, and hence effects of Einstein's special and general relativity theory need to be taken into account. The reduced gravitational influence in orbit, compared to the Earth's surface, causes a blue-shift and satellite clocks to tick a little faster, whereas time dilation, induced by the motion of the satellites, reduces clock frequency (Ashby, 2003). At higher orbit altitude, such as for GNSS clocks, the former effect prevails. Therefore, these atomic clocks are intentionally slowed down prior to the GNSS satellite's launch by a certain clock frequency shift, in order to compensate for their time acceleration in orbit (Ashby, 2014;

Mudrak et al., 2015). Additional periodically varying gravitational frequency shift and second-order Doppler shift, induced by the eccentricity of the satellite orbit, are modeled based on the LEO position $r_r$ and velocity $v_r$ at signal reception time and the GNSS position $r^t$ and velocity $v^t$ at signal transmission time, respectively (Kouba, 2015, 2004):

$$\delta t_{r,rel} = -2\frac{r_r \cdot v_r}{c^2}, \quad \delta t^t_{rel} = -2\frac{r^t \cdot v^t}{c^2}. \tag{6}$$

By convention, GNSS clock products provided by IGS and CODE analysis centers do not contain this so-called eccentricity

correction (Kouba, 2015) and for this reason it needs to be applied to the transmitter clock in RO excess phase processing. For LEO satellites, however, such a convention does not exist and the correction might be handled differently in POD processing. At WEGC, Bernese (v5.2) by default does not explicitly model this correction, whereas we had to modify NAPEOS to align the processing strategy between the two POD software packages. In practice, if the effect is not explicitly modeled it is absorbed in the POD clock error estimate and $\delta t_{r,rel}$ is set to zero. However, explicit modeling of the eccentricity correction of the LEO

clock can reduce variability and therefore slightly enhance interpolation residuals.

In addition to the relativistic effects acting on the satellite clocks we model the far smaller gravitational time delay acting on the GNSS signals transmitted between the GNSS and LEO satellites as follows:

$$\delta\rho^t_{r,rel} = \frac{2GM_E}{c^2} \quad ln\left(\frac{r^t + r_r + \rho^t_r}{r^t + r_r - \rho^t_r}\right), \tag{7}$$

where $G$ is the gravitational constant, $M_E$ is the mass of the Earth, and $r^t$ and $r_r$ are the satellite and receiver radial position at

signal transmission and reception times (Ashby, 2014; Mudrak et al., 2015). This effect is also referred to as the Shapiro time delay.

The GNSS and LEO orbits are provided with the satellite's mass center as reference point and hence need to be corrected for the antenna offsets (from COM) in order to reflect the proper locations of the RO signal's transmission and reception. In general, those antenna offsets are composed by the geometric antenna reference point (ARP), the electromagnetic phase center

offset (PCO) accounting for deviation from the geometric ARP, and, dependent on the signal's incoming direction, the phase center variation (PCV). As the PCV is a rather small effect, considering the short duration of an occultation event, it can be disregarded in RO processing (Hajj et al., 2002). The LEO RO antenna offsets (ARP and PCO) are applied using definitions





provided in the so-called ANTEX format (cf. Section 2.1) and common coordinate transformation from the satellite body frame to ECI (EUMETSAT, 2005). Corrections for the changing orientation of the satellite in space and the deviation from nominal
attitude during orbital revolution are not (yet) implemented in rOPS. However, for missions with stable orientation like Metop this correction is small. On the GNSS transmitter side, neither GNSS antenna offsets nor attitude, which have a far smaller effect on RO processing than the LEO antenna offsets, are modeled yet (Hunt et al., 2018).

After correct handling of the relativistic excentricity corrections and application of LEO antenna offsets, we can proceed and interpolate the LEO clock corrections, available at low 30-s sampling rate, to the high-rate (50 Hz) RO measurement time
stamps. This is done in an iterative procedure using a linear interpolation method until it converges at an empirically determined precision threshold. Once the corrected high-rate receive time is available, the LEO orbit and clock data from the POD-derived GSM-Geo orbit arcs are as well interpolated to these time stamps, using an 8-th order polynomial Lagrange interpolation for satellite positions and velocities and a linear interpolation for its clock biases.

While the reception time of RO signals is recorded by the receiver and its biases are modeled and estimated, the signal
transmission time of phase measurements needs to be reconstructed in order to allow extraction of corresponding GNSS orbit and clock data. Assuming a straight-line propagation in a vacuum medium of the signal is considered sufficient to infer the signal travel time $\tau$, elapsed during the signal's transmission at the GNSS satellite and the reception by RO receiver on the LEO satellite, with an iterative approach starting with $\tau_0 = 0$:

$$\tau_{r,n}^t = \frac{\rho_r^t}{c} = \frac{|r_r(t_r) - r^t(t_r - \tau_{r,n-1}^t)|}{c}, \text{ if } \tau_{r,n}^t - \tau_{r,n-1}^t < \text{threshold then abort, } n \in \{0,1,2,...\}. \tag{8}$$

and therefore the signal transmission time $t^t = t_r - \tau_r^t$. Within the signal travel time calculation routine, also the GNSS orbit positions are updated and interpolated to $t^t$. Once the iterative calculation of the signal transmission time converged, the GNSS velocity and clock data are interpolated the same way as performed for the LEO satellite. Note that the retrieved geometric distance $\rho_r^t$, required for the extraction of the RO excess phases, is a function of two time system variables $t^t$ and $t_r$.

### 3.1.4 Measurement Correction/Preparation & Differencing

At this point we have available all the correction terms necessary for the excess phase calculation in Equation 5: the geometric distance between corrected GNSS and LEO satellites at signal receive time $\rho_r^t(t_r)$, the interpolated clock bias estimates $\delta t_t(t^t)$, $\delta t_r(t_r)$, the corresponding relativistic eccentricity correction terms $\delta t_{r,rel}(t_r)$, $\delta t_{rel}^t(t^t)$ and the relativistic Shapiro effect $\delta t_{r,rel}^t(t_r)$. Effects not modeled, such as antenna phase wind-up (Bai et al., 2014, 2017), and residual errors are accounted for in the error budget $\epsilon$, which we will discuss in Section 3.2. However, depending on the RO receiver type, the raw measurements
recorded in CL, OL, or RS tracking are also subject to corrections and preparatory steps before the total phase $L_r^t(t_r)$ can be used in the calculation: the navigation data de-modulation; down-sampling to nominal measurement frequency (i.e., 50 Hz) of higher-sampled tropospheric measurements (e.g., Metop/GRAS 1 kHz RS); and assembling of the total phase.

GNSS signals are modulated with binary $[0, \pi]$ navigation data message (NDM) providing useful information, such as ephemeris, clock, ionospheric, and service parameters, but at the same time introducing carrier phase flips to the signal. In
CL tracking, these phase changes of $\pi$ are corrected in real-time at the RO receiver in orbit, whereas in lower troposphere





tracking (OL, RS) the NDM must be demodulated in post-processing on ground by either (Sokolovskiy et al., 2006): (1) internal removal by detection of phase switches between adjacent samples; (2) application of an externally provided NDM data stream (see Section 2.1; EUMETSAT (2018)). In general, and also at WEGC, the latter strategy is preferred, in particular for climate application, because increasing disturbances in the lower troposphere can exceed the NDM-induced phase flips and

introduce substantial retrieval errors if performing internal NDM removal (Sokolovskiy et al., 2009). Note that the previously calculated signal travel time $\tau$ is used to align the measurement data and the NDM bits at signal transmission time, which are then used directly to revert the phase flips in the measured residual I/Q components.

Measurement data exceeding the 50 Hz sampling rate for RO CL measurements are down-sampled to this nominal sampling frequency. Schreiner et al. (2011) investigated different down-sampling target rates for Metop/GRAS 1 kHz RS data and sug-

gested to use a sampling of 100 Hz for RO measurements in the lower troposphere but also 50 Hz data lead to closely similar results (Gorbunov et al., 2011). So far a sampling rate of 100 Hz has been realized for COSMIC-2 (Schreiner et al., 2020) and FengYun-3/GNOS (Bai et al., 2018). For GRAS RS data specifically, we calculate the arithmetic mean of 20 adjacent samples in order to obtain down-sampled RO signal at 50 Hz. The SNR is then recalculated from the raw I/Q components with the amplitude $A = \sqrt{I^2 + Q^2}$, while the NCO phase is interpolated linearly to 50 Hz.

With the corrected in-phase and quadrature I/Q signal components and the receiver model phase, the total measured phase can be constructed. First the $i$th sample of the residual phase in radians is calculated by $L_{rsd,i} = \text{atan}_2(Q_i, I_i)$ for all four quadrants, before unwrapping the signal so that adjacent samples of the residual phase are always within $\pi$:

$$L_{rsd,i} = L_{rsd,i} + 2k_i\pi \qquad \text{with} \quad k \in [-1, 0, 1], \tag{9}$$

where $k_i$ is defined by minimization of $|L_{rsd,i} - L_{rsd,i-1}|$. Finally, the total phase is obtained by $L_i = L_{nco,i} + \lambda \frac{1}{2\pi} L_{rsd,i}$,

where division by $2\pi$ converts the residual phase from radians to cycles and $\lambda$ denotes the wavelength of the GNSS signal in question.

With all terms available, we now can calculate the excess phase $\Delta L_r^t$ following Equation 5, representing a zero-differencing approach. This means that no clock differencing method is applied and the receiver clock bias estimates are employed directly in the differencing equation. This is the preferred option introducing lowest noise in the processing but is only practical for

RO missions with a sufficiently stable LEO clock (ultra-stable quartz oscillators; e.g., Metop, GRACE, FengYun-3). For other missions with less stable LEO clocks (e.g., CHAMP, COSMIC), the receiver clocks biases are removed by introducing a reference clock from a non-occulting GNSS satellite link in a so-called single differencing approach (Schreiner et al., 2010; Beyerle et al., 2005; Bai et al., 2018). This alternative differencing is not applied in this study.

### 3.2   Quality control and uncertainty estimation

Following the calculation of excess phase profiles as a function of time—including SLTP altitudes as initial vertical-levels variable—by the ODP-L1a *raw* processor, the quality and uncertainty of these profiles is assessed in the ODP-L1a *quality* processing. The ESM-Atm provides modeled excess phases for each measured RO event, which serve as an important input to



ODP-L1a *quality* (cf. Figure 2). Following the sequence we implemented, we will first discuss the quality assessment of the excess phase profiles, including possible rejection, before addressing the associated uncertainty estimation.

### 3.2.1 Quality control


The rOPS L1a quality control (QC) process flags, or (optionally) excludes, un-physical or low quality excess phase profiles from further processing in order to provide a standardized input for the uncertainty estimation and to ensure highest quality in the subsequent retrieval steps towards thermodynamic variables. If an event is not entirely rejected, specific QC flags are set (and written to a separate monitoring file) for data which are not passing the various quality checks or if certain parts of the

measurement are truncated at basically reliable top and bottom altitude levels. Key variables of the QC system are the observed excess phase profiles and the independent forward-model excess phase profiles as well as their delta-profiles with suitable low- and high-pass filtering applied.

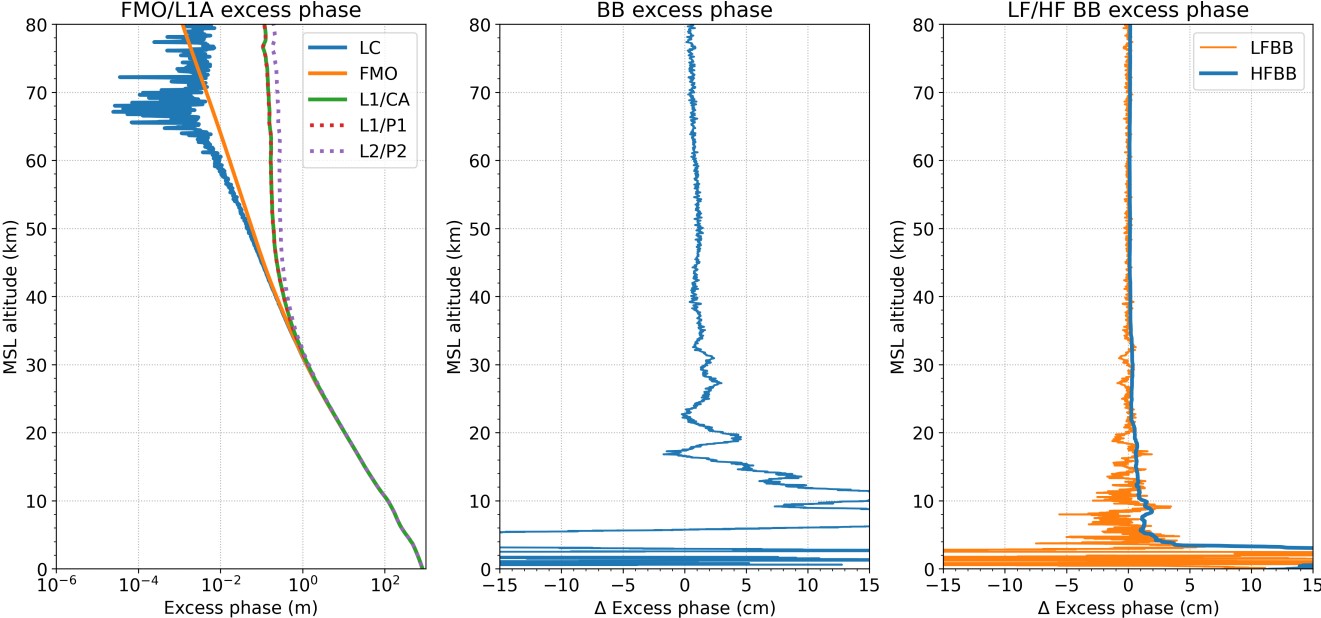

**Figure 7.** Illustration (left) of forward-modeled (FMO), observed (L1/CA, L1/P1, L2/P2), and linearly ionosphere-corrected (LC) excess phase profiles, all as function of (forward-modeled) MSL altitude. Additionally the derived baseband (BB) delta excess phase is shown (middle) and the lowpass-filtered (LF) BB as well as highpass-filtered (HF) BB delta excess phase profiles (right). These different phase profile examples are shown for an exemplary setting RO event from Metop-A (1 July 2008/00:01 UTC; lon: 37.6°; lat: -72.3°).

Central to the quality checks performed is the evaluation of the observed excess phase data against the independent forward-modeled data. Since the modeled excess phase profiles are free from ionospheric influences, we derive the ionosphere-corrected

excess phase profile by the classical first-order linear combination of the observed dual-frequency GNSS excess phases, e.g.,





for GPS (Schreiner et al., 2011):

$$L_c = L_1 + \frac{f_2^2}{f_1^2 - f_2^2}(L_1 - L_2), \tag{10}$$

with $f_1$ and $f_2$ being the GPS carrier frequencies of the corresponding $L_1$ and $L_2$ excess phase profiles. Figure 7 (left) illustrates such profiles.

The signal strength and thus the penetration depth in the troposphere differs among the different GNSS signals involved in Equation 10. Hence, the derived $L_c$ profile is restricted to the altitude range covered by the shorter of the two signals. To overcome this limitation we artificially extend the shorter signal using a linear extrapolation downwards to match the reach of the stronger signal. This is realized by determination of the linear gradient of the difference profile of both signals over a sufficiently long altitude range (10 km) above the end of the shorter signal, with the lowermost-allowed altitude of 15 km, to

avoid the usage of data with low SNR. The extrapolated part is then obtained by subtraction from the leading signal to obtain the extended weaker signal. The extended ionosphere-corrected excess phase profile $L_c$ in this form serves just as an auxiliary variable for the L1a QC. The ionospheric correction as part of RO retrieval is performed in a more advanced manner at the ODP-L1b bending angle level, including also higher-order correction (Danzer et al., 2021; Liu et al., 2020).

     Within rOPS, and specifically also for the ODP-L1a quality control, the analysis and processing on delta-profiles plays a

central role. Following the so-called *baseband approach*, with the subtraction of a background model profile $L_m$ (ESM-Atm) from the retrieved profile $L$ (ODP-L1a), biases from the (near-)exponentially varying RO profiles can be removed. This also leads to very small residual numerical errors of operators such as filters and derivatives: $\delta L_{BB} = L - L_m$.

     Residual noise in the high-frequency domain of the baseband excess phase $\delta L_{BB}$ can be suppressed by a lowpass filter, leading to a delta-phase profile without noise and smaller variations, which better displays the large scale effects of the atmosphere.

We use a Blackman-Windowed-Sinc (BWS, Smith (1999)) lowpass-filter $\boldsymbol{F}^{BWS}$ with a cutoff frequency $f_c$ at 0.5 Hz. With a sampling rate of $f_s = 50$ Hz the suitable window size is $2 \cdot f_s/f_c + 1 = 201$ data points, which corresponds to an effective filtering window size of 100 points or 2 s. For a more detailed description and discussion of advantages of a BWS filter over moving-average Boxcar filters see (Schwarz et al., 2018, Appendix 1.2 therein). The lowpass-filtered baseband excess phase is then obtained as: $\delta L_{LFBB} = \boldsymbol{F}^{BWS}\delta L_{BB}$.

Subtraction of the lowpass-filtered baseband excess phase from the baseband excess phase leads to a highpass-filtered baseband excess phase: $\delta L_{HFBB} = \delta L_{BB} - \delta L_{LFBB}$. This double-delta profile is virtually unbiased down to the lowest altitudes and contains almost only information about the noise level, which allows for inspection of the noise level, and of potential outliers, of the individual excess phase profiles.

     Based on these derived delta-phase profiles (cf. Figure 7), the quality check procedure is summarized in Table 4, providing

a concise overview on the main quality checks performed. Note that parameters listed are tailored for Metop/GRAS data and might require some modification for other RO missions.

     The QC process starts from raw excess phase data (raw signal) with some basic data preparation steps and initial checks for physical plausibility of the measurement. Initially, all the profiles are cropped to a minimum of $z_{SLTP}^{min} = $ -250 km and a maximum of $z_{SLTP}^{max} = 90$ km altitude, respectively. Subsequently, the measurements, recorded at irregular measurement times,



**Table 4.** Overview on the rOPS L1a quality processing. All parameters apply to Metop/GRAS data. Middle column $f_{QC}$ provides rejection fractions for every single quality control step for all data (9 months) processed by WEGC for this study.

| task | $\mathbf{f_{QC}}$ | description |
|---|---|---|
| background check | 0.15 % | check if corresponding background file exists |
| crop profiles | N/A | cut profiles at minimum $z_{\mathrm{SLTP}}^{\min}$ (-250 km) and maximum $z_{\mathrm{SLTP}}^{\max}$ (90 km) SLTP altitude |
| equidistant grid | N/A | interpolate irregular measurement time stamps from $L$ to the strict 50 Hz stamps from $L_m$ |
| sampling check | 0 % | check consistency and rate drift of interpolated excess phase $L$ |
| altitude check | 1.96 % | check coverage of minimum required height range from 23 km to 70 km SLTP altitude |
| normalization | N/A | basic normalization to median phase between 60 and 70 km SLTP altitude |
| raw phase check | 0.28 % | check baseband delta excess phase $\delta L_{\mathrm{BB}}$ to be within $\pm 500$ m |
| outlier check | 0 % | check outliers based on $\delta L_{\mathrm{BB}}$ using $\pm 5\sigma$ bounds |
| stddev check | 1.60 % | check standard deviation range of baseband excess phase $\delta L_{c,\mathrm{BB}}$ (top) and $\delta L_{c,\mathrm{HFBB}}$ (bottom) |
| bounds check | 0.18 % | check bounds of delta-phase based on $\delta L_{c,HFBB}$ |
| smoothness check | 0 % | check delta-phase gradient based on $\delta L_{c,HFBB}$ |

are interpolated to a strict equidistant time grid of 50 Hz based on the model grid. Additionally, the sampling rate consistency and drift are checked to be within the plausible maximum limits of 0.015 s and $10^{-5}$ s/min, respectively, over the entire cropped altitude range. The interpolated and cropped excess phase profiles are written to the output used for further L1b processing; all subsequent steps solely serve the quality control and are stored as interim output.

For further processing, profiles which do not span over a minimum altitude range from $z_{\mathrm{SLTP}}^{\mathrm{bot}} = 23$ km to $z_{\mathrm{SLTP}}^{\mathrm{top}} = 70$ km

are dismissed or optionally flagged. In a next step the entire profile is normalized to the median value between 60 km and 70 km SLTP altitude. In order to eliminate obviously un-physical profile behaviors, the delta-profile $\delta L_{\mathrm{BB}}$ for any signal profile is not allowed to exceed 500 m (an empirically determined value for sorting out gross outliers) over the minimum SLTP altitude range defined above.

Highly important to the QC, and for safeguarding the further processing, is the outlier detection performed on the baseband

delta excess phase $\delta L_{\mathrm{BB}}$ for each signal profile separately. This baseband delta excess phase allows for very robust outlier check, since the forward-modeled excess phase is a noise-free reference, in order to focus on potential irregular spikes and patterns in the data itself. The outlier detection criteria are based on a moving median-referenced percentile-based detection calculation. The 16th and 84th percentile are used for an estimation of the standard deviation ($\pm\sigma$). A data point is considered an outlier if it is greater than the five-fold value of this standard deviation around the median; if more than 3 % outlier values

are detected across the profile, the entire profile gets rejected.

Since values are not distributed symmetrically around the median, an adequate window size of $2 \cdot f_s = 100$ data points was found as result of sensitivity tests that evaluated statistics and counted the number of profiles with at least one detected





outlier, while employing different window sizes (Seidl, 2018). At this stage of the processing, outliers are only detected, while in the early part of the ODP-L1b processing outliers are then (statistically) corrected, because further calculations including

averages and standard deviations respond very sensitively to outliers. That correction replaces any detected outlier value by a statistically reasonable value randomly drawn from a normal distribution with standard deviation $\sigma = \frac{1}{2}(p_{84} - p_{16})$ within the interval $[-3\sigma, +3\sigma]$.

Next, inspection of the moving standard deviation of delta excess phase profiles, using a window size of $2 \cdot f_s = 100$ data points to estimate it, helps to estimate reliable top and bottom altitude levels, within which the quality of the excess phase data

is considered sufficiently reliable to retrieve sufficiently accurate variables in the subsequent geometric-optics bending angle retrieval. For the determination of a reliable top altitude level in the mesosphere, we apply a moving standard deviation to the baseband delta excess phase $\delta Lc_{\mathrm{BB}}$ and check where a threshold of 3 cm (an empirically determined reasonable value) is exceeded for the first time when searching upwards.

For bottom altitude level determination, the standard deviation of the highpass-filtered baseband delta excess phase $\delta L_{\mathrm{HFBB}}$

of L1, L2 and LC profiles are checked, now searching downwards, for the maximum of either a fixed value (3 cm) or a relative value (0.1 %), the latter relative to the forward-modeled excess phase $L_{\mathrm{m}}$. If the moving standard deviation thresholds are found to be exceeded already within the minimum-required altitude region from $z_{\mathrm{SLTP}}^{\mathrm{bot}}$ to $z_{\mathrm{SLTP}}^{\mathrm{top}}$ then the respected profile (and whole RO event) is either discarded or flagged. The top and bottom altitude levels determined are stored for later use, such as for optionally cutting the profiles to their best-quality range before further retrieval steps.

A final pair of fundamental QC checks for the physical plausibility of a processed excess phase profile by checking bounds and smoothness of delta excess phase profiles of $L_c$. The bounds are checked on the baseband delta excess phase $\delta Lc_{BB}$ with defined limits derived from sensitivity studies and using general physical knowledge for expected magnitudes in different atmospheric altitude regions. Specifically, every data point of $\delta Lc_{BB}$ is checked to be within a threshold of 15 cm in the mesosphere from 50 km upwards and the maximum $\max[30\,\mathrm{cm}, 0.01 \cdot L_{\mathrm{m}}]$ for altitudes from 30 km downwards, respectively.

For altitudes between (upper stratosphere), the bound is defined by linearly increasing between the lower stratospheric and mesospheric bounds, i.e., from 30 cm to 15 cm (Seidl, 2018).

The smoothness check is based on a 5-point derivative of the highpass-filtered baseband delta excess phase $Lc_{\mathrm{HFBB}}$, yielding a delta-phase derivative profile used for detection of rapidly fluctuating noise or spikes with un-physical magnitude. The smoothness bound is set such that no point in the derivative profile shall exceed 7.5 m/s. Additionally, it is cross-checked to

fall within a relative limit of 75% of the 5-point derivative of $L_{\mathrm{m}}$. Profiles failing the bounds check and/or smoothness check within the minimal-required altitude range are discarded.

### 3.2.2 Characterization of excess phase uncertainty

Following the QC of the RO events initially processed by the ODP-L1a processor (raw), the uncertainty of the excess phase profiles found of adequate quality (i.e., passing the QC) is characterized and empirically estimated as the next step of the

processing. This new approach to assess the uncertainty of each individual RO event is conducted following the Guide to the Expression of Uncertainty in Measurement (GUM; JCGM2008, JCGM2012) and its proposed handling and terminology of



uncertainties. Overall the GUM is considered the standard guideline for the uncertainty estimation and propagation in rOPS, as already realized in parts of the retrieval (Innerkofler et al., 2020; Schwarz et al., 2017, 2018; Schwarz, 2018; Li et al., 2019).

As summarized by Schwarz et al. (2018) we categorize uncertainties into estimated random uncertainties and estimated systematic uncertainties, the latter with differentiation between basic and apparent systematic uncertainties. Effects included in the estimated random uncertainty budget are of unpredictable or stochastic temporal and spatial variability in repeated observations. These effects are essentially stationary in a statistical sense so that we can estimate their statistics also from individual RO event data, given the high noise-resolving sampling rate along the vertical profiling.

Systematic effects (biases), which can not be quantified using statistical data analysis based on just one individual RO pro-
file, are estimated and corrected for if known, as recommended by the GUM. The remaining residual biases are assumed to stay within a (conservative) bound estimate, which we refer to as estimated systematic uncertainty. Depending on their nature, we distinguish two types: (1) estimated basic systematic uncertainties which appear and will remain systematic despite of averaging over individual RO events and (2) estimated apparent systematic uncertainties which will essentially behave as random uncertainties in ensemble averaging over many RO events. It is important to distinguish these two subtypes, since the former
will not average out and therefore fundamentally limit the (absolute) accuracy of ensemble averages such as climatologies.

With respect to the phase observations (Equation 4) a total uncertainty budget for the GNSS excess phase measurements can be approached as:

$$u_{\text{tot}} = \sqrt{u_{\text{atm}}^2 + u_{\text{ion}}^2 + u_{\text{r,clk}}^2 + u_{\text{t,clk}}^2 + u_{\text{vel}}^2 + u_{\text{therm}}^2 + u_{\text{mult}}^2 + u_{\text{cycle}}^2 + u_{\text{phase}}^2}, \tag{11}$$

where $u_{\text{tot}}$ denotes the total uncertainty assigned to the excess phase profile, $u_{\text{atm}}$ the uncertainty inferred from the neutral
atmosphere influences, $u_{\text{ion}}$ the uncertainty inferred from the ionospheric influences, $u_{\text{r,clk}}$ the receiver clock uncertainty, $u_{\text{t,clk}}$ the transmitter clock uncertainty, $u_{\text{vel}}$ the velocity drift uncertainty, $u_{\text{therm}}$ the thermal noise, $u_{\text{mult}}$ uncertainty due to local multipath, and $u_{\text{phase}}$ the phase noise comprising residual error influences.

Remaining error sources, such as phase wind-up and tide and ocean loading effects, which are not considered in Equation 11, are expected to be below the order of $10^{-6}$ in relative errors and are therefore disregarded in the uncertainty budget. Also
interpolation and smoothing of data introduces errors in the calculations, e.g., by disregarding terms of higher order in an approximation or through the imperfect representation of numerical values. Within the excess phase processing these errors are mostly attributable to the interpolation of the receiver and transmitter orbits and clocks. The magnitude of these errors was empirically determined and found to be smaller than the order of $10^{-5}$ in relative errors, and hence is also considered negligible compared to the other components in the error budget.

The uncertainties of the transmitter and receiver clocks are modeled based on their one-second Allan deviation, using typical values for atomic clocks and quartz clocks of GNSS and LEO satellites, respectively (cf. Section 3.1.3). The relative frequency error for white frequency noise averaged over the sampling time is simply simulated as a random number between $-1$ and 1 (Harting, 1996). To achieve the correct one-second Allan deviation of the final calculated clock noise, the one-second Allan Deviation has to be corrected for the used sampling rate. The relative frequency error for the sampling period becomes





therefore:

$$y_n = \sqrt{6}\, A_{1sec}\, rand_n\, \frac{c}{\sqrt{\tau_s}}, \tag{12}$$

where $y_n$ is the relative frequency error averaged over the sampling period, $\tau_s$, $A_{1sec}$ is the one-second Allan deviation, $rand_n$ is the random number between $-1$ and $1$, and $c$ the speed of light in vacuum. For spaceborne single differencing, a flicker frequency noise is added to the clock uncertainty.

Thermal noise denotes the errors due to the random movements of electrons in the electronic components of a GNSS receiver. For carrier phase signals of nominal strength, the thermal noise in the carrier-tracking phase-locked loop (PLL) of a GNSS receiver can be modeled as

$$\sigma_{PLL} = \frac{\lambda}{2\pi}\sqrt{\frac{B_L}{C/N_0}}, \tag{13}$$

where $B_L$ is the carrier loop noise bandwidth, $C/N_0$ is the carrier-to-noise-power-density ratio, and $\lambda$ is the carrier phase
wavelength. In a realistic approximation, with $C/N_0 = 45$ dB-Hz and $B_L = 2$ Hz, the thermal noise for the GPS L1 signal is 0.2 millimeter (Langley, 1997). Accounting for the weaker GNSS signals and non-optimal conditions in the lower atmosphere, where also open-loop tracking is used, we safely assume a conservative bound for the influence of the thermal noise on phase measurements of 1 mm for all measured GNSS signals.

The random velocity error, estimated along with the POD processing (Section 2.2.1), also contributes to the total uncertainty
budget. It introduces a linear drift to the measurement and is therefore categorized as estimated apparent systematic uncertainty. Assuming a random velocity uncertainty of $v_{u_r}^r = 0.02$ mm/s of the receiver satellite orbit, following

$$u_{vel} = v_{u_r}^r \cdot (|t_{top} - t_{bot}|), \tag{14}$$

the total uncertainty introduced by the velocity drift uncertainty $u_{vel}$ would account for 1.2 mm after a typical RO event duration of 60 seconds.

As stated in Section 3.1.1, phase ambiguities do not pose a problem in the excess phase processing since constant terms can be eliminated from the observation equation. However, undetected cycle slips can introduce a phase shift of half or full carrier wave cycles in order of a several centimeters, depending on the wavelength of the GNSS frequency. With decreasing altitude the ratio between the absolute value of the excess phase and the magnitude of possible cycle slips is increasing and additionally propagation media effects in the troposphere require a delay-locked loop (DLL) measurement mode, which makes
cycle slips occur more frequently and left undetected. Therefore, to account for these undetected cycle slips as an estimated basic uncertainty, we include a coverage factor $c = 0.001$ m/min which increases linearly with decreasing altitude for DLL measurements:

$$u_{cycle} = c \cdot (|t_{top}^{DLL} - t_{bot}^{DLL}|). \tag{15}$$

Local multipath occurs when the incoming signals at the receiver satellite are reflected and scattered before reaching the
vicinity of the GNSS antenna and thus signals with different paths are detected simultaneously at the receiver. The effect



depends on the spacecraft geometry, the occultation viewing geometry, and the electrical properties in the vicinity of the receiver antenna (Kursinski et al., 1997). The possible phase shifts of up to a few centimeters, introduced by local multipath, can be reduced by modeling the effect and the use of directional antennas. Additionally, local multipath effects can be reduced by dismissing incoming left-hand polarized signals, since the right-hand circular-polarized GNSS signals are left-hand polarized

after reflection at the surface of the LEO. The residual local multipath error effects on the phase measurements are modeled using a sinusoidal model, for representative broad beam antennas used in GNSS RO (Steiner and Kirchengast, 2005). The sinusoidal shaped function is defined with a multipath phase error amplitude of 0.5 mm and period set to 60 seconds, resulting in multipath errors up to 1 mm.

The influence of the propagation media (neutral atmosphere and ionosphere) itself cannot be quantified directly because

ionospheric scintillations and atmospheric turbulences lead to variations in the properties of the atmosphere which can be hardly captured by models. However, if we assume that the defined uncertainty budget in Equation (11) includes all possible major error sources, the uncertainty arising from the neutral atmosphere and ionosphere can be inferred from an empirical estimate of the total random uncertainty and subtraction of the terms categorized as contributors to the random uncertainty:

$$u_{\text{atm+ion}}^2 = u_{\text{atm}}^2 + u_{\text{ion}}^2 = u_{\text{emp}}^2 - u_{\text{r, clk}}^2 - u_{\text{t, clk}}^2 - u_{\text{thermal}}^2. \tag{16}$$

The empirical uncertainty estimate is determined based on the double-delta excess phase profiles $\delta L_{\text{HFBB}}$ smoothed with a moving standard deviation with a window size of $2 \cdot f_s + 1 = 101$, like we already used in in the QC processing (Section 3.2.1).

As final output, the combined estimated random, basic systematic, and apparent systematic uncertainties are appended to the L1a output, while the individual uncertainty estimates are stored in a "monitoring file" together with the QC output. Figure 8 illustrates the daily component-wise and combined uncertainty estimates. The overall estimated excess phase uncertainties

serve then as input to the subsequent uncertainty propagation in the ODP-L1b bending angle retrieval (Schwarz et al., 2018). An evaluation of the uncertainty estimation can be found in the results Section 4.3.







**Figure 8.** Overview of exemplary estimated random $u_r$, basic systematic $u_b$, and apparent systematic $u_a$ uncertainty components, respectively, from rOPS ODP-L1a processing for Metop-A on July 1, 2008. For each component, individual estimates for each of the 781 events (thin grey profiles) are shown together with the daily median uncertainty profile (highlighted thick orange).





## 4 Results and discussion

In this section we evaluate a set of three 3-monthly periods of RO data at excess phase level: the summers 2008 and 2013 (both July-August-September, JAS) as well as the winter 2019/2020 (December-January-February, DJF), for the Metop-A/B/C RO
data as available given their launch times. Starting with a sensitivity analysis of the excess phases computed by the rOPS, we analyzed how the measured profiles compare against forward-modeled profiles, how different orbit solution input influences the L1a excess phase processing, and how the evaluated differences of collocated profiles agree between the three Metop missions and their GRAS receivers.

Subsequently, we present a statistical ensemble evaluation of WEGC's rOPS data in comparison to the excess phase data
processed and provided by EUMETSAT and UCAR. The last subsection discusses results related to uncertainty estimation of the excess phase profiles processed at WEGC.

The number of profiles for each study period per Metop satellite is depicted in Figure 9. All data are available in rOPS-L1a file format, where the external excess phase data from EUMETSAT and UCAR underwent a basic conversion routine. Overall, in average, the daily number of profiles amounts to 647 profiles (Metop-A), 610 profiles (Metop-B), and 559 profiles (Metop-
C), with reduced numbers in the later periods. In the following evaluation, differences between observations of rising or setting RO events or differences from geographic regions (zonal bands) are discussed whenever worth to take into account.

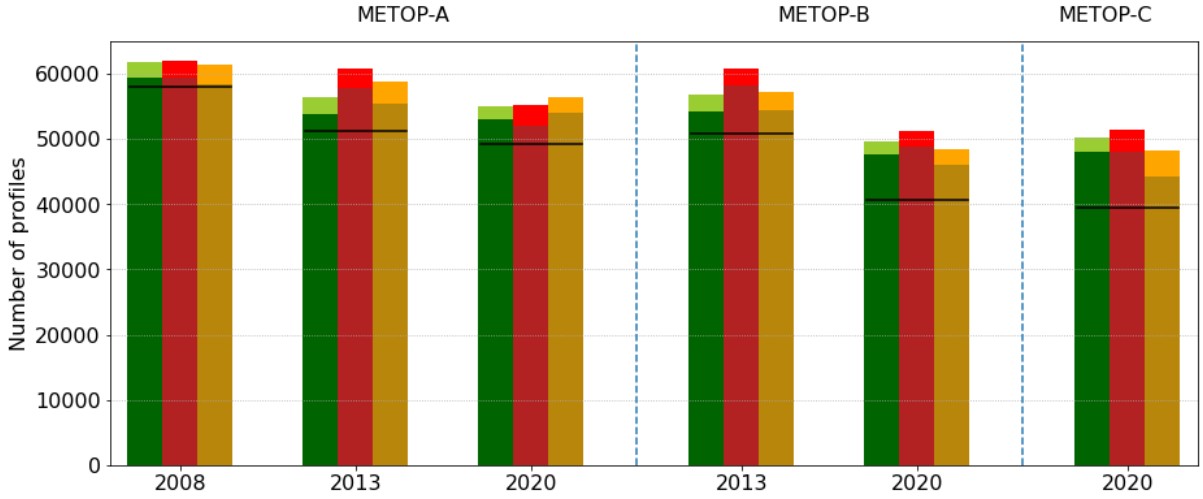

**Figure 9.** Number of RO events per time period 2008-JAS, 2013-JAS, 2020-DJF processed at WEGC (green), and converted from EUMET-SAT (red) and CDAAC (orange) to rOPS format. Darker shaded area of the bars indicate profiles, which successfully passed quality control. The black line indicates the number of profiles matched by event identifier and common to all three processing centers.

### 4.1 Sensitivity analysis

In order to evaluate the quality of processed excess phases and their sensitivity to certain effects we considered their difference to selected reanalysis data, their sensitivity to different orbit inputs and different measurement geometries, as well as their




dependency on the specific Metop satellite. For a comprehensive and well comparative analysis we focused on three vertical atmospheric regions and their layer averages:

- middle *troposphere* (3-7 km): this atmospheric region in the lowermost section of the atmosphere was selected for representation of the (moist) middle to lower troposphere. With the restriction to above 3 km, the boundary layer and its special behavior that requires separate study (e.g., Gorbunov2021, Gorbunov2018, Zeng2016) was disregarded.

- lower *stratosphere* (20-25 km): representing the lower stratosphere core-strength region of RO soundings, with narrow uncertainty and minimal influence of background data or of any tropospheric or tropopause effects in the retrieval.

- lower *mesosphere* (50-60 km): representing the lower mesosphere with small absolute excess phases of order 10 cm left only but still without major ionospheric influences that start higher. The layer thickness of this region is increased compared to the tropospheric and stratospheric layers for increasing averaging in view of the less dense atmosphere at

these higher altitudes.

### 4.1.1   Atmospheric background modeled profiles: O-B

The measured RO data, processed to excess phase level by WEGC's rOPS, are compared to forward-modeled excess phase profiles from ECMWF ERA5 reanalyses (Section 2.2.3). Since per construction there is a corresponding reference profile extracted from the highly-resolved ECMWF data for each observed RO event, the number of profile validation pairs is high.

Statistical evaluation of WEGC processing against the forward-modeled atmospheric background profiles is shown in Figure 10. In general the median of the intercomparison is seen at millimetric level for all atmospheric regions and Metop satellites considered. The spread of the data increases due to higher variability and model imperfections at lower altitudes in the troposphere, where also a portion of skewed data can be seen. Differences between rising and setting RO events and differences in geographic distribution (not shown) are found to be very small with median differences of a few millimeter.

### 680   4.1.2   Orbit sensitivity

Various studies have documented that the accuracy of the orbit, and in particular the velocities and estimated clock biases, co-determine the quality of the derived excess phase profiles (e.g., Kursinski et al. (1997); Montenbruck et al. (2008)). For this reason, WEGC implemented a novel system routinely evaluating the quality of the calculated orbits which was introduced by Innerkofler et al. (2020). In this study we conducted the WEGC L1a excess phase processing based on 3 different orbit

solutions, from WEGC, EUMETSAT, and UCAR, respectively.

The comparison results of these independently computed orbit solutions are summarized in Table 5. These results focus on 2013-JAS and 2020-DJF, since data from 2008 were not available from EUMETSAT for this study part. In order to ensure a stable orbit geometry and excess phase comparison, one generally aims for a daily 3D-RMS threshold of 5 cm in position and 0.05 mm/s in velocity (Innerkofler et al., 2020). This is generally satisfied for the WEGC orbit solutions in comparison to

EUMETSAT, whereas UCAR exhibits slightly higher deviations as part of this intercomparison.





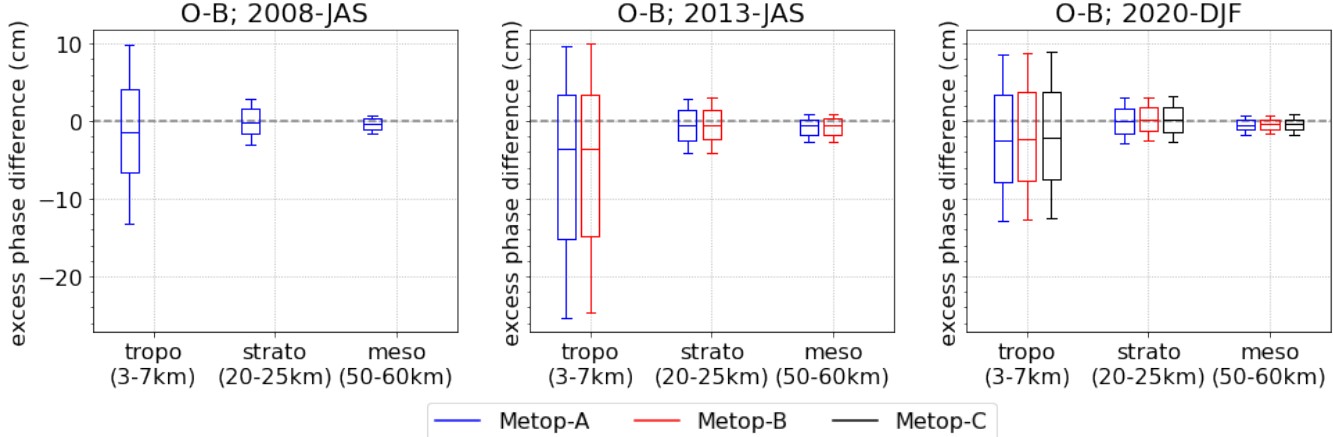

**Figure 10.** Sensitivity analysis results from comparing WEGC rOPS L1a excess phase profiles from Metop-A (blue), Metop-B (red), and Metop-C (black) against collocated forward-modeled profiles from ERA5, for the three 3-month periods (2008, 2013, 2020; left to right) and three representative altitude layers (left to right within each panel), with the box-whiskers depicting the median (mid-line), 16% to 84% range (box), and 5% to 95% range (whiskers).

Comparison results of rOPS processed excess phases based on WEGC-Bernese and orbit processing against EUMETSAT POD exhibit a similar agreement across all vertical layers, time periods and satellites. As depicted in Figure 11, the median of the comparison experiences a slight positive bias and is enclosed by a stable percentile envelope, which is only slowly increasing when moving towards lower atmospheric regions. Overall, the 5% to 95% percentile range stays below about
1 cm. Setting events show a tendency, at millimetric level, towards smaller averages and rising events towards larger averages, compared to the total median statistic.

### 4.1.3   Sensitivity of sensors

In order to investigate the influence and difference between the very closely similar receivers and satellite platforms, profiles of
the three Metop satellites were co-located and binned together, before also calculating these differences for statistical inspection. Though the satellites are essentially identical in construction, different firmware, shift of orbital planes, different satellite lifetimes, as well as degradation of instruments and hardware may influence the differences between profiles of different satellites.

For the evaluation of platform dependent differences the collocation algorithm was restricted to match profiles within 2
hours in time and 200 km in space. In addition to this, we limited the analysis to the period where at least the first two satellites launched, Metop-A and Metop-B, were in space already.



**Table 5.** Metop orbit intercomparisons for position and velocity in terms of 3D-RMS for different orbit solutions for 2013-JAS and 2020-DJF periods, compared to the EUMETSAT-Napeos POD solution used as reference. Days flagged to exceed 20 cm in 3D-RMS (about 3 days per satellite and 3-month period) have been excluded from the statistics.

| ID | 2013-JAS | | 2020-DJF | |
|---|---|---|---|---|
| | **position** (cm) | **velocity** (mm/s) | **position** (cm) | **velocity** (mm/s) |
| **Metop-A:** vs. EUMETSAT-Napeos | | | | |
| WEGC-Bernese | $3.86 \pm 0.37$ | $0.036 \pm 0.004$ | $5.22 \pm 1.84$ | $0.049 \pm 0.021$ |
| WEGC-Napeos | $4.22 \pm 0.49$ | $0.031 \pm 0.005$ | $4.55 \pm 1.62$ | $0.039 \pm 0.016$ |
| UCAR-Bernese | $5.80 \pm 0.47$ | $0.067 \pm 0.004$ | $8.31 \pm 2.18$ | $0.088 \pm 0.021$ |
| **Metop-B:** vs. EUMETSAT-Napeos | | | | |
| WEGC-Bernese | $3.67 \pm 0.26$ | $0.040 \pm 0.004$ | $5.19 \pm 1.74$ | $0.053 \pm 0.015$ |
| WEGC-Napeos | $3.95 \pm 0.52$ | $0.032 \pm 0.006$ | $4.93 \pm 1.58$ | $0.044 \pm 0.016$ |
| UCAR-Bernese | $5.57 \pm 0.55$ | $0.062 \pm 0.005$ | $9.81 \pm 2.02$ | $0.108 \pm 0.022$ |
| **Metop-C:** vs. EUMETSAT-Napeos | | | | |
| WEGC-Bernese | - | - | $5.12 \pm 1.17$ | $0.089 \pm 0.026$ |
| WEGC-Napeos | - | - | $4.46 \pm 0.91$ | $0.048 \pm 0.011$ |
| UCAR-Bernese | - | - | $8.78 \pm 3.20$ | $0.038 \pm 0.011$ |

Figure 12 shows that the resulting differences are generally larger compared to strictly collocated profile-to-profile comparisons, particularly in the troposphere, where the space-time-mislocations of the RO event locations grossly dominate the (quasi-random) differences. However, the results exhibit a stable median, which is well centered at zero (at cm-level consistency) for all altitude layers. Only for Metop-C in winter 2020, a small positive deviation from zero is visible for the tropospheric layer, which due to the high overall variability cannot be interpreted in more detail based on these statistics.

## 4.2 Statistical analysis

In order to evaluate the performance and ensure highest quality profiles by the new rOPS L1a processing, the excess phase profiles calculated at WEGC were evaluated also against profiles from EUMETSAT (von Engeln et al., 2009) and UCAR CDAAC (Schreiner et al., 2011), the latter serving as additional independent data source. Prior to comparison and the calculation of profile differences, the external L1a profiles from EUMETSAT and UCAR were converted to rOPS-L1a format, including the calculation of consistent unique event information and some format alignments of the datasets (Section 2.2), with the help of which the RO events from the three different processing centers were reliably matched.

For thorough analysis we separated the profiles into a combined mesospheric and stratospheric section, using MSL altitude (mapped using the corresponding forward-model profiles) as vertical coordinate from 10 km to 80 km. In this range, absolute

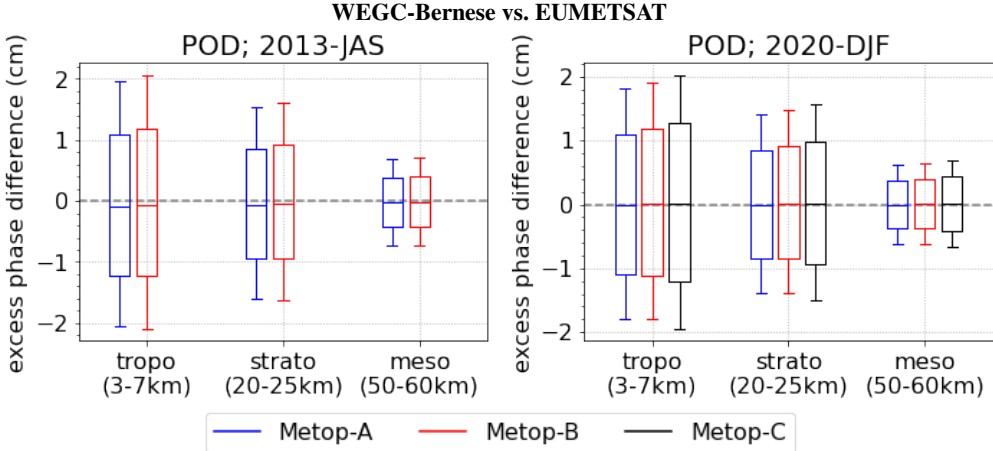

**Figure 11.** Sensitivity analysis results from comparing WEGC rOPS L1a excess phase profiles from Metop-A (blue), Metop-B (red), and Metop-C (black) against profiles based on different POD solutions, for the two 3-month periods (2013, 2020; left to right) and three representative altitude layers (left to right within each panel), with the box-whiskers depicting the median (mid-line), 16% to 84% range (box), and 5% to 95% range (whiskers).

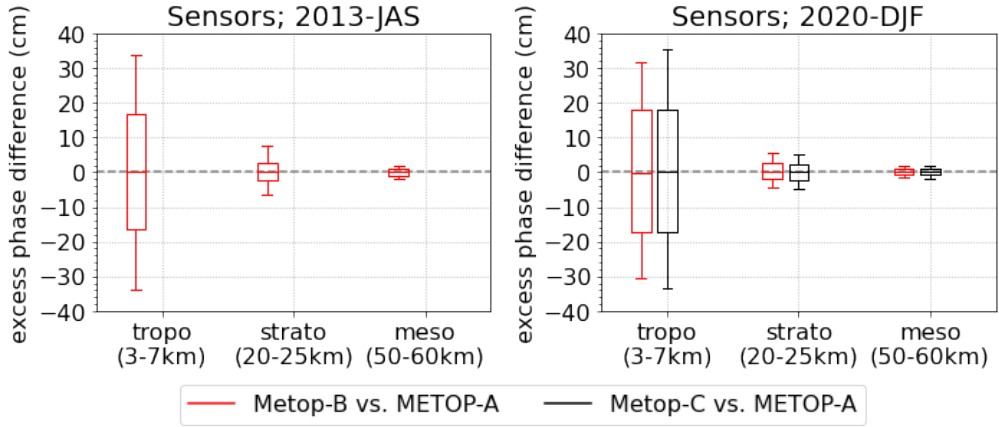

**Figure 12.** Sensitivity analysis results from comparing co-located WEGC rOPS L1a excess phase profiles from Metop-B (red) and Metop-C (black) against Metop-A, selected as representative reference, for two 3-month periods (2013, 2020: left to right) and three representative altitude layers (left to right within each panel), with the box-whiskers depicting the median (mid-line), 16% to 84% range (box) and 5% to 95% range (whiskers).

phase and differences are comparatively small. Second, a tropospheric region using SLTP altitudes from $0\,\mathrm{km}$ to $-100\,\mathrm{km}$ was defined, wherein loss of lock of the tracking usually occurs and where open loop tracking introduces larger differences. Considering a standard atmosphere model, this approximately equals an MSL altitude from about 13 km down to the last few hundred meters above Earth's surface within the boundary layer.





Although several profiles reach down to $-250\,km$ SLTP altitude, the number of profiles significantly decreases below about $-100\,km$ SLTP altitude, which is why we focused the tropospheric statistics down to this level. Furthermore, strong refraction, possibly super-refraction, and other disturbances complicate an overall statistical comparison at the very low altitude levels below $-100$ km SLTP altitude.

Figure 13 shows the number of available matched profiles from WEGC and CDAAC versus EUMETSAT for 2008-JAS,
2013-JAS, and 2020-DJF, for all RO events from Metop-A/B/C and additionally separated for rising and setting events. In comparison, generally the number of profile differences from setting events, compared to rising events, is higher for all time periods and also show an overall deeper penetration.

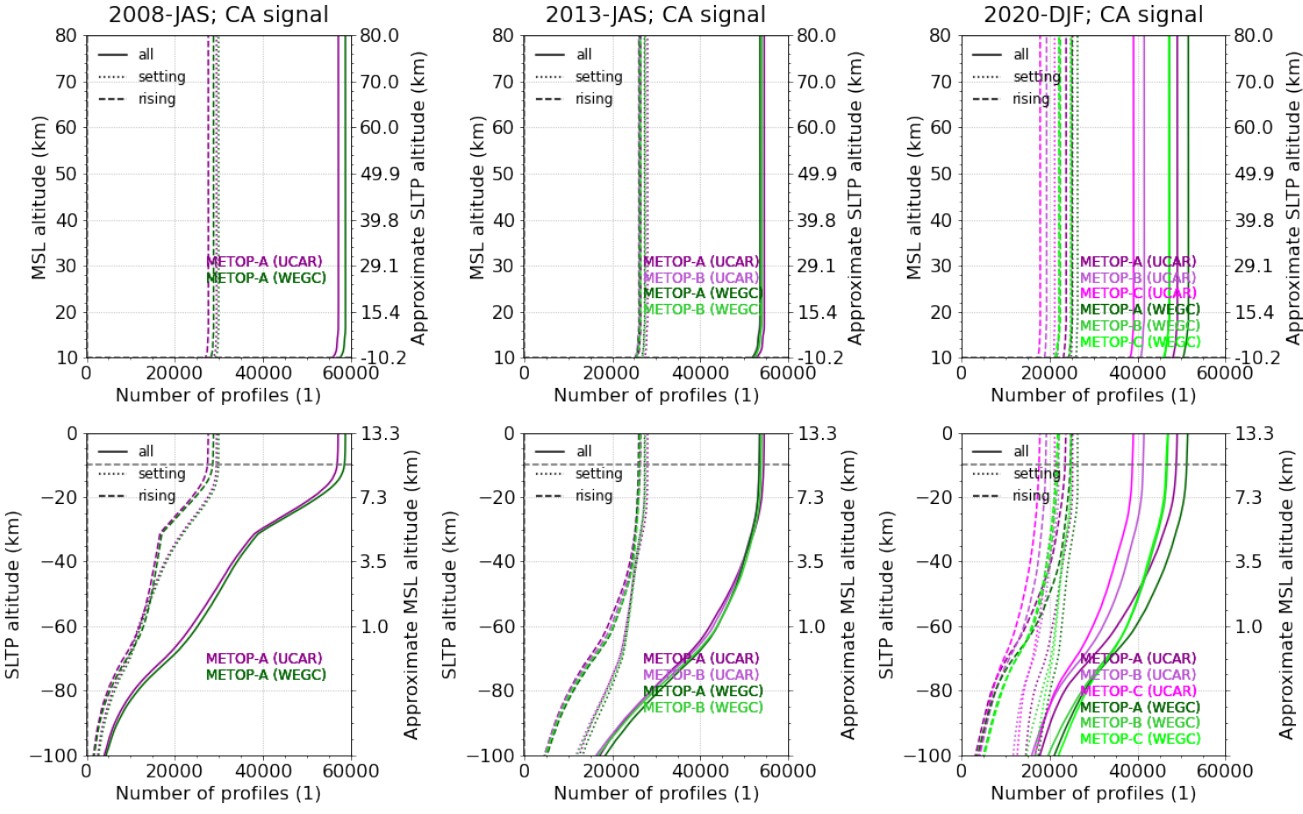

**Figure 13.** Number of excess phase difference-pair profiles used for the ensemble statistics for WEGC vs. EUMETSAT (green) and WEGC vs. UCAR (purple), with focus on the stratosphere and mesosphere region (top row) and the troposphere region (bottom row). Difference-pair profile numbers are shown for L1/CA excess phase profiles, featuring deepest penetration and being most relevant for the subsequent L1b retrieval.





The inter-center comparison results at excess phase level between WEGC and UCAR against EUMETSAT (the latter used as

reference) are shown in Figure 14 for the stratosphere/mesosphere-focused region and in Figure 15 for the troposphere-focused region, respectively.

A closer look at these 3-monthly excess phase difference statistics as a function of MSL altitude (see Figure 14) shows good agreement and a robust difference statistics. The systematic difference between WEGC and EUMETSAT shows a virtually bias-free consistency (differences within 1 mm), apart from the 2013 period, where a salient kink can be observed at around

11 km MSL altitude. More detailed investigation showed that this behavior occurs for all time periods considered, although the altitude at which it can be observed is slightly below 10 km MSL altitude for the other time periods and hence outside the focus region shown in Figure 14.

This characteristic can be attributed to a weakness in the transition from closed-loop to open-loop data for rising events in the WEGC processing, a matter which is currently under refinement. Systematic differences between UCAR and EUMETSAT

show a positive bias, up to 2 mm to 3 mm, around the tropopause, which is most pronounced for setting events. Interestingly, in 2020-DJF the biases for rising events prevail. With increasing excess phase magnitude, with decreasing height, the standard deviation of the comparison of WEGC with EUMETSAT also gradually increases, from 80 km down to 10 km in a linear and fairly symmetrical manner. The standard deviation of the differences corresponds to an envelope of about ± 6 mm and ± 8 mm (somewhat increased in 2020), demonstrating millimetric-level consistency of the random deviations. With UCAR as

candidate dataset, slightly increased standard deviations in the order of 1 cm can be observed near an altitude of 10 km.

The statistical intercomparison in the troposphere, depicted in Figure 15, shows a good agreement between WEGC and EUMETSAT excess phase profiles, with an almost bias-free comparison down to about −80 km SLTP altitude and a slightly increased systematic difference below. Separate investigation of rising and setting events reveals increased systematic differences starting already at about −50 km for rising events. However, the contribution to the overall statistics is limited because

of the lower number of rising profile differences in that altitude region (cf. Figure 13).

For the intercomparison statistics of UCAR against EUMETSAT, increased systematic differences are found starting at about −40 km in 2013-JAS and 2020-DJF, a characteristic only found below about −80 km in 2008. The biases observed are similar for all Metop satellites and most pronounced for setting events in 2020-DJF. At about −100 km, their magnitude is in the order of about 4 times the signal's wavelength, while rising events feature smaller but positive biases.

Although these characteristics arise in the (deep) boundary layer, where proper signal tracking is challenging, the salient deviations between UCAR excess phases and EUMETSAT and WEGC processing results point to remaining issues in the different processing approaches. However, such further investigation is beyond the scope of this study and will be addressed in follow-on intercomparison work.

The standard deviation of the comparison statistics shows an increase of the spread of the data with the onset at about −20

km SLTP for the comparisons from WEGC and UCAR against EUMETSAT, along with a decreasing number of difference-pair profiles. For settings events considered separately, however, the standard deviation for WEGC against EUMETSAT is narrower and starts to increase from −50 km SLTP altitude downwards. In general, the standard deviation amounts to about ± 80 cm in





most cases at $-100$ km SLTP altitude, with increased values for UCAR compared to WEGC as candidate dataset in the lowest hundreds-of-meters of the boundary layer close to the Earth's surface.

**Figure 14.** Systematic difference (median) and standard deviation profiles from the statistical ensemble differences between excess phase profiles from WEGC (green) and UCAR/CDAAC (magenta) compared to EUMETSAT. All events (left column), setting events (middle column), and rising events (right column) are shown for 2008-JAS (top row) to 2020-DJF (bottom row). The altitude range focuses on $10-80$ km MSL altitude.





**Figure 15.** Systematic differences (median) and standard deviation from the statistical ensemble differences between excess phase profiles from WEGC (green) and UCAR/CDAAC (magenta) compared to EUMETSAT. All events (left column), setting events (middle column), and rising events (right column) are shown for 2008-JAS (top row) to 2020-DJF (bottom row). The altitude range focuses on -100−0 km SLTP altitude.





## 4.3 Uncertainty analysis

As described in Section 3.2.2, the empirical uncertainty estimation at excess phase level delivers estimated random uncertainty and, separately, estimated basic and apparent systematic uncertainties for each individual RO event. Related to this Figure 16

provides, as an synthesis result, a concise overview of daily mean estimates of these uncertainties, for a representative example day of each of the three 3-month periods investigated.

In the mesosphere and upper stratosphere, the estimated random and systematic uncertainty components are small at millimetric level, while towards lower altitudes, where the absolute excess phase increases rapidly, the estimated random uncertainty component amounts to several centimeters (which is still very small in relative terms, though). The estimated systematic un-

certainty component is seen very small at millimetric level, and slightly increases with decreasing altitude to reach near 0.5 cm in the deep troposphere.

Overall, the daily averages of the uncertainty estimates as depicted in Figure 16 show similar characteristics for all three time periods and for all the three different Metop satellites.

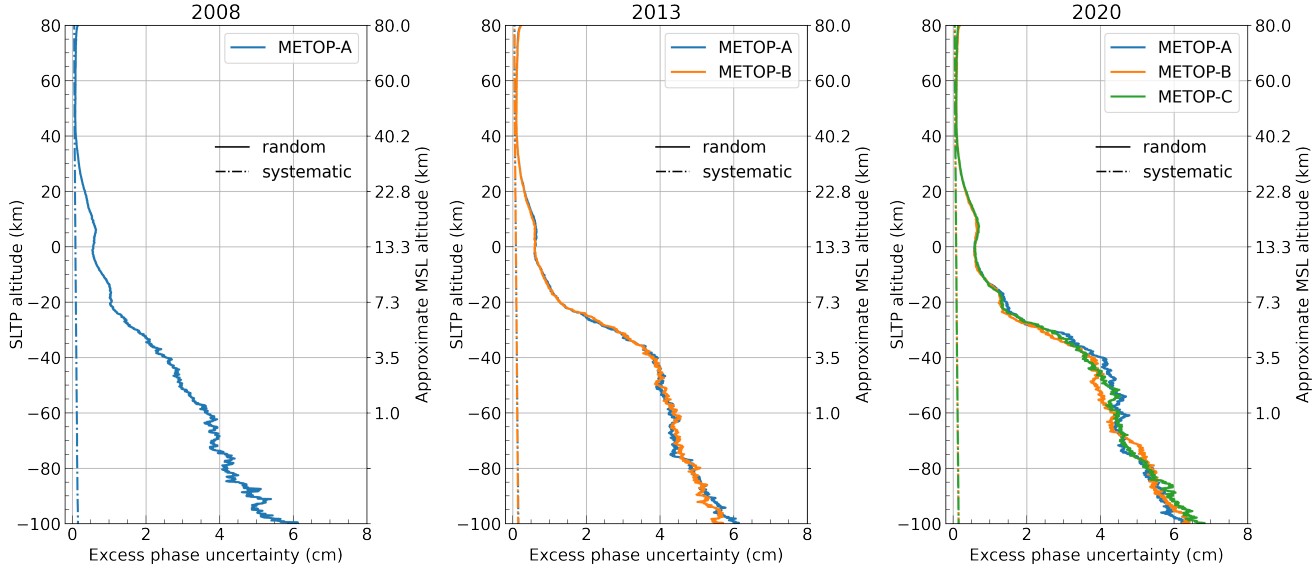

**Figure 16.** Daily averages (median) of estimated random and estimated systematic (combined basic and apparent) uncertainties based on individual-RO-event uncertainty profiles for the example days of July 1, 2008 (Metop-A; left), July 1, 2013 (Metop-A/B; middle), and January 2, 2020 (Metop-A/B/C; right).



## 5   Summary & conclusions

The derivation of climate benchmark data from RO measurements requires a rigorous processing from the raw GNSS signal occultation measurements (including their georeferencing in space-time) to essential climate variables (ECVs) such as temperature, with narrow uncertainties. In order to encompass the entire RO data processing, the new rOPS processing system at WEGC has recently seen major advancements in Level 1a (L1a) processing, with the integration of the POD processing (Innerkofler et al., 2020) as well as the excess phase processing including uncertainty estimation that was presented and evaluated

in this study.

The rOPS L1a excess phase processing, for itself algorithmically concise, is complemented by integrated quality control and uncertainty estimation, which requires an advanced framework of various subsystems, which we introduced step-by-step in Section 2. Starting with the input data preparation we discussed the various input data sources. Building on the POD processing, integrated in the rOPS, the DSM (daily system modeling) subsystem delivers daily LEO receiver and GNSS

transmitter satellites orbit and clock data. The observation geometry setup computed by the ESM (event system modeling) facilitates the provision of basic event-based geometry and environmental context information. Complemented by atmospheric profiles modeling, the ESM also provides reliable forward-modeled excess phase profiles based on collocated (re-)analysis data such as from the European Reanalysis ERA5. These forward-modeled profiles, including a forward mapping of mean-seal-level (MSL) altitudes to the RO event's time grid, are found highly valuable for the quality control and for supporting the

uncertainty estimation.

After the introduction of the rOPS L1a processing framework, we provided an in-depth description of the main processing algorithms for the derivation of excess phase profiles in addition to a due account of the GNSS observables and corresponding receiver tracking modes. Subsequently, an elaborated quality control, to a major extent supported by forward-modeled excess phase profiles, removes or alternatively flags profiles of insufficient or degraded quality. The primary addition, however, is the

component-wise estimation of excess phase uncertainty profiles for each individual RO event, which serve as starting point for the uncertainty propagation in the subsequent RO retrieval chain down to the atmospheric ECV profiles.

The validation of the new processing was based on three 3-month periods (in 2008, 2013, 2020) for achieving a robust statistical analysis. In a sensitivity analysis for inspecting the quality and reliability of the resulting excess phases, we focused on three atmospheric layers: lower mesosphere (50-60 km), lower stratosphere (20-25 km), and middle troposphere (3-7 km);

investigating differences between observed and collocated ERA5-derived profiles, the influence of different orbit and clock inputs, as well as cross-platform comparison between profiles from the different Metop satellites.

The comparison against forward-modeled profiles revealed small systematic differences mostly of millimetric level only, based on the multi-monthly statistics. The exchange of WEGC LEO orbit solutions with orbit solutions from EUMETSAT or UCAR (all of adequate quality within about 5 cm in position and 0.05 mm/s in velocity in an orbit inter-comparison) in the

rOPS processing led to negligible differences in the results for the different time periods and Metop satellites. Intercomparing matched observations between different Metop satellites, using collocation criteria of 2 hours in time and 200 km in space, revealed larger differences dominated by the space-time differences of the RO event locations, but still the median-difference





results, found at millimeter level deviation, confirmed an overall statistically very tight consistency across the different Metop receiver platforms.

An inter-center comparison of WEGC- and UCAR-processed excess phases against EUMETSAT-processed data focused on an upper (10 km - 80 km MSL altitudes) and lower (0 km to $-100$ km SLTP altitude) atmospheric region, respectively. While this analysis revealed some small discrepancies in the processing of rising RO events at WEGC, the excess phases processed with rOPS show better agreement to EUMETSAT excess phases than to UCAR ones. In the stratosphere/mesosphere-focused region, the differences between WEGC and EUMETSAT amount to $1 \pm 6$ mm, compared to $3 \pm 8$ mm for UCAR against

EUMETSAT. In the troposphere-focused region, differences stay small down to about $-80$ km SLTP altitude, with a standard deviation increase to near $\pm 80$ cm, still corresponding to very small relative differences. Larger deviations of UCAR excess phases at lower altitudes suggest noticeable differences compared to the processing strategies at WEGC and EUMETSAT.

    A sensitivity check of the stability of the uncertainty processing, based on daily averages of example days from the study time periods, showed closely consistent systematic and random uncertainty estimates across all time periods and the three Metop

satellites. Both the estimated random and estimated systematic uncertainties stay at (sub-)millimetric level in the stratosphere and mesosphere, while at tropospheric altitudes in the dense atmosphere with large excess phases, where signal tracking is challenged, the random uncertainty estimates increase up to several centimeters.

    In follow-on work we will investigate the long-time evolution of the uncertainty budget and compare the characteristics to those of other RO missions. The approach of the uncertainty estimations of various components at excess phase level is

also subject to repeated improvements and refinements. Overall the results provide evidence for the rOPS L1a processing's capability to provide valuable uncertainty estimates for further processing, supported by a rigorous quality control for climate-quality data processing demands. Demonstrating the delivery of robust and accurate excess phase profiles, the results indicate that the rOPS L1a subsystem is a reliable new excess phase processor, which is versatile also in the sense that it can handle multiple different input data sources, such as orbit and excess phase data also from other processing centers.

While this study focused on a thorough introduction and careful initial evaluation based on Metop-A/B/C satellite data, the rOPS L1a processing is set up as a multi-mission system capable to process RO measurements also from CHAMP, GRACE, COSMIC, COSMIC-2, FY-3, and Spire-RO satellite series. In this way it will support the provision of long-term multi-mission RO data records including uncertainty estimation, for the benefit of climate monitoring and research.

*Code availability.* The code used to produce the results of this study is available from the corresponding author upon qualified request.

*Data availability.* The (numeric) data underlying the results of this study are available from the corresponding author upon qualified request.

*Author contributions.* Conceptualization and methodology was provided by JI and GK, embedded into the Reference Occultation Processing System (rOPS) framework conceived by GK. JI elaborated the detailed algorithms, performed the computational implementation and the analysis, prepared the figures, and wrote the first draft of the paper. GK supported this work with the provision of detailed design input and advice on all aspects of the study and significant contributions to writing of the paper. The computational implementation was extensively



supported by MS. MS, CM, and YA advised on a range of technical and scientific aspects and contributed through writing-review and editing. All authors participated in the consolidation of the paper for submission and towards publication.

*Competing interests.*  The authors declare that they have no conflict of interest.

*Acknowledgements.*  We thank EUMETSAT Darmstadt and UCAR/CDAAC Boulder for access to their RO excess phase and orbit data and ECMWF Reading for access to their analysis and forecast data. We also thank V. Proschek from WEGC for technical support and

valuable discussions and D. Hunt and B. Schreiner from UCAR for valuable advice. The work was funded by the Austrian Aeronautics and Space Agency of the Austrian Research Promotion Agency (FFG-ALR) under the Austrian Space Applications Programme (ASAP) projects ATROMSAF1 (project no. 859771) and ATROMSAF2 (project no. 873696) funded by the Ministry of Climate, Energy, Environment, Mobility, Innovation, and Technology (BMK). Funding was also provided by the Austrian Science Fund (FWF) under research grant W1256-G15 (Doctoral Programme Climate Change – Uncertainties, Thresholds and Coping Strategies).





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
