# Peer review of "GNSS radio occultation excess phase processing for climate applications including uncertainty estimation"

_Atmospheric Measurement Techniques, 2023_

## Referee Comment (RC2)

Review for "GNSS radio occultation excess phase processing for climate applications including uncertainty estimation":

This paper "GNSS radio occultation excess phase processing for climate applications including uncertainty estimation" by Innerkofler et al. describes a new RO excess phase processing system including excess phase uncertainty estimation for Metop series satellites. The excess phase profiles derived with such system are compared against those from different processing centers with different POD and excess phase algorithms. The uncertainty estimation of excess phase thus can be helpful to trace the excess phase errors back to SI standard. The main purpose of such "reprocessing" of operational missions is to provide climate quality data records. First there indeed exist the needs in RO community for reprocessing of Metop A/B/C excess phase datasets for inter-center comparison so that the structural uncertainties in the dry temperature can be traced back to excess phase or observational level. There are CDR products from EUMETSAT ROM SAF, but their RO products are based on existing excess phase profiles from other processing centers. UCAR CDAAC also has different versions of RO datasets for the same missions. Second, determining the uncertainty of RO bending angle retrievals are often limited to local spectral width (LSW), which is hard to connect with the excess phase uncertainty. Thus this study, the excess phase processing with uncertainty estimation, is scientifically important and a significant contribution to GNSS RO community, not only because it can be used for quality control for excess phase profiles, but also because the excess phase uncertainties can be further possibly quantified to derive the bending angle uncertainties. Technically, the processing system uses improved GNSS/Leo POD solutions, follows the standard excess phase processing procedure adopted by other missions/centers but with rigorous quality control. It uses the zero-differencing clock bias removal algorithm which depends on the ultra-stable clock onboard the Metop-Satellites. The RO excess phase processing algorithm description is solid and covers all the related aspects. The quality control relies on the excess phase modeling, including geometric and atmospheric modeling, which is a significant step in excess phase algorithm. Overall, the excess phase processing of three three-months periods for Metop A/B/C shows successful reprocessing of Metop RO excess phase profiles, with uncertainty estimation, and within expected differences compared with datasets from other centers. The system can be used to generate a long time series for climate applications. Overall, this paper is well written and organized, with technical description in details, and presents the results clearly. The logic of the study is scientifically sound. The excess phase uncertainty estimation can be applied to other missions. I recommended this paper to be published at AMT after some minor revision.

I am little bit concerned the large difference in excess phase in lower atmosphere (Figure 15) from UCAR. What's the main reason caused the large difference in standard deviation/biases? Is this related to how the excess phase model and/or the filtering/smoothing algorithm used for excess phase processing? The author should explain that in depth. Is this a proof that the actual uncertainties maybe larger than proposed (e.g: the STD compared with UCAR at 3.5 km MSL is more than 40cm, but the uncertainty in excess phase is less than 4cm.)?

Though the excess phase uncertainty estimation is important, how this uncertainty can be translated into the Doppler shift and then into bending angle is not clearly mentioned. Some discussion on how

the excess phase uncertainty propagates further into bending angle should be given for the cases given. After all, the bending angle or derived temperature products are the Essential Climate Variable.

Technically, the excess phase processing in this study seem ignored both the GNSS and LEO satellite attitude information, please explain in detail how this can affect the error budget in excess phase.

Figure 15, please explain why the STD profiles in figure 15 (e.g. left dotted line and right dotted line in any subfigure) are not symmetric even systematic bias approaches zero?

Minor Comments:

Line 63, Is excess phase measurement accuracy/uncertainty really SI traceable given the excess phase model used, the GPS bit time series used, and the cycle slip correction uncertainty in the lower atmosphere?

Figure 2, Are the attitude data belong to the auxiliary datasets from IERS? Aren't they provided by the mission operation?

Line 106, In table I, I believe the LEO attitude is important and should be labeled. It looks the usage of the LEO attitude is optional. But how could you convert the antenna offsets from space body frame to ECI without attitude/quaternion information?

Line 130, does this reconstruction include the POD phase/pseudo-range also (RINEX files)?

Line 138, I believe UCAR have a more specific data address (url) to point to the exact location of the datasets used.

Line 205, this has puzzled me. I think this is different from that used by UCAR. At the mean event time, the straight line may not be tangent to the WGS-84 ellipsoid surface. Unless the tangent point can be defined first and then the time difference (really small though) can be neglected.  Please explain. How sensible the different profile location definition can affect the excess phase quality control (atmospheric modeling), especially at lower atmosphere?

Line 240, 'is used T and  ln(p)', should be 'is used for T and ln(p)'? It would be better to give a reference for this interpolation scheme or a reason why these schemes are the best (e.g. ROPP manual compares different interpolation schemes). Different interpolation scheme can certainly affect how the bending angle bias look alike in different altitude.

Line 260, the angle $\zeta$ should be between the velocity vector and the leo position vector, please label correctly in Figure. 3.

Line 285, Carries phase can't be called phase pseudo-range, since the time measurements and phase measurements use different mechanisms in GNSS positioning techniques. It would be more appropriate to use its name 'carrier phase' than 'phase pseudo range.'

Line 337, Rewrite Eq. 5 to be consistent with Eq. 4, such as the ionosphere correction has opposite sign with atmospheric delay term, missed C in the third item in Eq. 5, inconsistent sign (+/-) between receiver/transmitter clock bias correction, etc.

Line 346-349, some terms are misleading. You do not need to correct the clock bias, but to remove it. the antenna offset needs to be calculated in the proper coordinate system and added to the mass center of the satellites. The distance is not between satellites, but between receiver and transmitter antennas (pcvs/or offset since pcvs may not be used here). Please rewrite accurately.

Figure 6 and from other context, why the LEO/GNSS satellites attitude input is optional? Are all the antennas offsets/pcvs defined in such a way that the attitude is not needed? How does this affect the error budget and excess phase itself? If attitude are not needed, you may also need to explicitly explain how the antenna offsets are applied.

Line 395-397, I do not understand the sentence ' common coordinate transformation from satellite body frame to ECI', isn't this the satellite attitude information (usually given as quaternions).

Line 410: Here the time delay correction does not consider the GNSS antenna offset. There is neglected time bias of about 2/C (assume GNSS antenna offset length of 2m) about 7 ns. Please justify how this affects the excess phase calculation with zero-differencing methods. For single differencing, this may be absorbed by differencing itself.

Line 436-437, down sampling of RS data to 50HZ, the authors used the 20 samples arithmetic mean. Please explain how the 20 samples arithmetic mean affect the cycle slips (if not corrected yet) especially for lower atmosphere.

Line 453, the Metop POD antenna are not designed to track high rate GNSS signals thus the single differencing may not be effectively used (as COSMIC does) with low rate POD antenna observations.

Line 505, Aren't the sampling rate defined at the receiver time with a constant interval? Please explain what caused the drift.

Line 549, what's the criteria to use 7.5 m/s.

Line 614, DLL already defined at line 310.

Line 619, please define the $t_{bot}^{Dll}$ and $t_{top}^{DLL}$. This looks quite small. Given one minute of travel time in the lower atmosphere, the cycle slip error is only 0.001m=1mm? If this is true, how do you explain large excess phase difference in lower troposphere between different processing centers?

Line 655, Why does the Metop-C have less daily RO profiles than Metop-A/B?

Line 731-732, it is hard to follow what the authors talk about. Are they trying to compare the setting and rising differences in total profile numbers or to compare inter-center difference?

---

## Author Comment (AC1)

**Response to Referee #1**

*(**Referee report**: https://doi.org/10.5194/amt-2023-28-RC1)*

**Manuscript:**

Innerkofler, J., Kirchengast, G., Schwärz, M., Marquardt, C., and Andres, Y.: *GNSS radio occultation excess phase processing for climate applications including uncertainty estimation*, Atmos. Meas. Tech. Discuss. [preprint], https://doi.org/10.5194/amt-2023-28, in review, 2023.
* * *
The authors thank the reviewer very much for the constructive and detailed feedback to the manuscript. We thoroughly considered all comments and carefully revised the manuscript accounting for them. Below are our point-by-point responses.

Comments by the reviewer are cited black upright, our responses are red. Line numbers used in our responses refer to the original AMT Discussions paper and text updates in the revised manuscript are quoted below in blue)

All citations referenced are provided in the revised manuscript.
* * *
**Summary**

The authors present a detailed and valuable summary of the rOPS L1a excess phase processing system that includes excess phase uncertainty estimation. The detailed processing description is well done. The uncertainty estimate portion of the paper should be updated to address the questions/comments below.

One aspect that the paper does not include and should, that would improve it, would be to include a discussion in the intro of how the uncertainty estimates described here would be used to improve the quality of the final ECVs. Are these uncertainties being used now to better derive BA, N, ...? Or will they only be propagated to higher level to provide uncertainty estimates of the ECVs? What about vertically correlated phase errors, or phase rates, due to LEO orbits, GNSS clocks or ionosphere residuals? Are they taken into account? These correlations will lead to larger errors for ECVs when integrated downward with the Abel inversion.

The subsequent usage of the estimated uncertainties at excess phase level and their impact on the quality of the following processing is discussed in the references provided at L69-L73. The uncertainties are used in the upper boundary initialization (statistical optimization) of bending angles before the refractivity retrieval as well as in the moist air profiles retrieval as part of the backgroud information (in the 1DVar/optimal estimation). We added the following paragraph to provide the reader with this information and advise how uncertainties are handled within the subsequent retrieval chain:

The random and systematic uncertainty estimates at excess phase level are then propagated through the entire ODP retrieval chain in order to provide the final ECVs with their associated uncertainties. Additionally, the uncertainties quantified are employed in part of the retrieval operators of rOPS to improve the derivation of variables (e.g., ionosphere correction, statistical optimization, moist air retrieval). For details on the uncertainty propagation along this chain, starting from the estimates at excess phase level, see Schwarz et al. (2018, 2017); Schwarz (2018); Li et al. (2019).

Another concern is the 3 data periods used in the study – they do not cover the most challenging regimes that RO has to track in. Analyzing a more challenging period may shed more light on RO error sources and uncertainty and better inform the community.

Regarding the selection of the test data periods comments are provided below (first comment under "Detailed Questions/Comments").

I believe this paper is a valuable contribution to RO and climate research and should be published with minor revisions.

The authors thank the reviewer very much for the valuable comments that helped to significantly improve the manuscript.

**Details Questions/Comments**

L121: 3 3-month time periods:

- 2008 (Jul-Sep), solar min
- 2013 (Jul-Sep), solar max, but not in Equatorial plasma bubble scintillation season
- 2019-20 (Dec-Feb), solar min

These 3 periods do not present the strongest challenge for RO data processing, quality control, and uncertainty evaluation. Solar max periods during Sept-March present the greatest challenge for RO especially in the equatorial region. This paper would provide a better understanding of RO uncertainty and issues if it included the most challenging conditions for RO during solar max and Fall-spring periods. If the authors can't include analysis of these data, then they should at least discuss any issues seen in these more challenging data.

Thank you for pointing towards the more challenging test period in the 2013 equatorial plasma bubble season for the propagation of radio signals. After careful consideration however, we came to the conclusion that the test data periods chosen serve the demonstrative purpose of this introductory paper of the L1a excess phase processing including uncertainty estimation and that reprocessing would be beyond the scope of this purpose. However, we will keep the referee's comment regarding the special case of equatorial plasma bubble season in mind for follow-up evaluations.

L146: ERA5 analysis used for validation of RO profiles. This validation is tricky, since ERA5 analysis already assimilates the RO profiles.

Yes, as stated in L150, the ERA5 analysis assimilates observational data including RO measurements. In this study the use of the ERA5 analysis data is limited to the sensitivity analysis performed in Section 4.1.1. For the task of pure assessment of the quality and stability of the implemented excess phase processing we consider this sufficient, although we are aware that the assimilation of RO data has an influence on the analysis (e.g., Schmidt, 2008). For genuine validation studies, the use of independent datasets is clearly advisable, in addition to the use of the ERA5 analysis data. We added a change as follows:

"… while the ERA5 analysis data are used for the provision of reference profiles in part of the sensitivity analysis of the RO retrieval."

L152: use suit, not suite

Thank you, the typo has been corrected in the manuscript.

L164: What if you don't have the GNSS navigation bits from the ground network? Do you process it anyway?

If GNSS navigation bit data are not available from the ground network we apply an internal removal by detection of phase switches between adjacent samples. Therefore, we process the data in any case. A more detailed description of navigation bit handling is provided in the paragraph starting at L423.

L175: use builds, not builts

Thank you, corrected in the manuscript.

L181: use pseudo-range, not pseudocode

Thank you, corrected in the manuscript.

L185: Can you provide some information in the text about the magnitudes of the computed systematic and random orbit uncertainties?

The following information regarding the estimated orbit uncertainties from the precise orbit determination is now provided in the revised manuscript:

"In general, the combined-3D position and velocity uncertainties estimates for the Metop satellite series amount to about 1.9 cm and 0.02 mm/s (for random) and 5.0 cm and 0.05 mm/s (for systematic), respectively."

L200: Table 2, should the units of GNSS clock bias be seconds (not m)?

Yes correct; changed in the manuscript.

L242: Does the linear combination of L1 and L2 phase at the same time result in over estimation of uncertainty. I assume it does since the L1 and L2 ray paths are different.

In the estimation of the uncertainty the linearly-combined excess phase profile LC from of L1 and L2 data includes ionospheric residuals, while the modeled excess phase is based on the neutral gas atmosphere without consideration of ionospheric influences. This leads to a conservative estimation of the uncertainty, as assumed by the reviewer. The LC excess phase profile here is only used for auxiliary information; in the state retrieval, the ionosphere correction takes place at bending angle level (as described in Schwarz et al., 2018).

L270: Q: how good is your model excess phase profile? Why not raytrace through ERA5 to get the model excess phase profile? I guess this won't work in the lower troposphere, but I doubt the excess phase data in the LT are very useful. You could at least validate your model excess phase above the LT where signal is small with the raytracing excess phase.

Evaluation with raytracing showed that the difference is very small, apart from the lower troposphere (where still the relative difference to data is very small; see, e.g., Schwarz et al., 2018; Appendix example therein on time series filtering based on the delta-signal upon subtracting the modeled excess phase).

L304: use, occultations that may miss ….

Thank you, changed in the manuscript.

L310-315: One disadvantage of the GRAS OL approach that should be mentioned here is that since it must track the CA code, it loses lock when the CA signal gets too noisy in the LT which results in some data gaps. Data gaps are bad for RO and can break SI-traceability of the observations. What do you do about data gaps?

In order to minimize the effect of data gaps on the SI-traceability of the observations, we analyze the measured data segments per individual occultation event and treat them following recommendations of the data provider as follows. We select the longest continuous closed-loop (CL) and raw-sampling (RS) data segments not allowing sampling-time deviations within these segments to be larger than 5 % of the

nominal sampling time stamps. The RS data are then down-sampled to 50 Hz and termed open-loop (OL) data thereafter. The two selected CL and OL data segments are usually overlapping or adjacent. When combining the CL and OL data we dismissed the OL if there was any gap between the two data segments. This accordingly reduces the tropospheric penetration depth for those specific occultation events. In the latest implementation of the rOPS, following the data provider recommendations, a maximum gap of 1.2 s between CL and OL is allowed, which is bridged by linear interpolation. SI-traceability is somewhat degraded by how efficient the data gaps are connected. We noted this in the manuscript as follows:

"This can lead to data gaps when the C/A code tracking loses lock caused by challenging tracking conditions in the lower troposphere (Schreiner et al., 2011). Therefore, in order not to degrade SI-traceability, we restrict the processed data to the longest continuous CL and RS data segments not allowing any gaps between these two data segments. Another … "

L359: remove 'most'. I don't think most RO missions use USO's like Metop, i.e., C-1, Spire, Champ, Kompsat-5, ..

Since there are quite a few RO missions besides Metop who use USOs (e.g., GRACE, FY-3, C-2) we rephrased the sentence accordingly.

"However, among other RO missions, Metop-A/B/C use ultra-stable quartz oscillators that are likewise highly accurate over the short term of RO events."

L461: A problem with this is that if you apply a conservative QC before uncertainty estimation, your uncertainty estimates will be under-estimated. Can you add some discussion on how you came up with the proper QC to obtain reliable uncertainty estimates?

In general, we aim to only exclude un-physical profiles from the subsequent uncertainty estimation and the subsequent ODP-retrieval, in order to facilitate properly estimated uncertainty (without influence of illegitimate outlier data). This is also done in order to ensure inclusion of all reasonable profiles for rigorous climate processing.

L465-470: Figure 7, is the BB, just the LC – FMO?

Also, Fig7 right panel, shouldn't the HPBB have the high frequency variations, and the LPBB have the low frequency variations?

Exactly the BB is LC - FMO, this is also described in more detail in lines 484 – 487. The HFBB actually is the difference between BB and LFBB (cf. line 495), which reduces the high-frequency variations of HFBB compared to LFBB.

L477: use, shorter weak signal

Thank you, changed in the manuscript.

L517: If an outlier is detected, do you remove it so there is a gap in the data, or do you replace it with an interpolated value? Also, what if your model profile is far from truth, how can you be sure you are not removing accurate data? If you remove outliers and create gaps, how can you be sure that you maintain SI-traceability throughout the profile?

Within the level 1a excess phase processing outliers are only detected. Later they are (statistically) corrected in the level 1b bending angle retrieval (with adjusted uncertainties). Using a statistically sound method for the outlier detection algorithm reduces the chance of removing accurate data. For this reason, and in order to maintain SI-traceability, outliers are corrected following this statistical approach in order to prevent data gaps. If an outlier is detected then it is replaced by a normally distributed random number drawn within a $\pm 3\sigma$ standard deviation range. It is hence an adequate fill value not distorting the statistics.

L525: What happens with the outlier detection when there is a cycle slip? L1 or L2? A cycle slip will look like a step in the BB excess phase. Also, it is assumed that occultations with large small-scale ionosphere residuals are captured with your QC algorithm. Which QC check best catches these ionospherically disturbed profiles? You showed combined QC %'s for all 3 data periods. Did you notice any differences in the QC %'s for the 3 different data periods? I would expect there to be a higher percentage of QC'd profiles for the solar max period.

Will have to do a separated calculation of the percentages for the different periods to answer this question.

The outlier detection checks only within the altitude range, where the altitude mapping of the forward modeling has converged. If a cycle slip occurs within this altitude range (and has not been corrected within the cycle slip detection algorithm before) the resulting step might cause the profile to be flagged (falsely) accordingly. Actually, classical scintillations are too small to be captured in the QC. We recalculated the QC rejection rates for the 3 different data periods and could not find major differences between the periods in 2008, 2013, and 2020.

L563: use, sampling rate along the vertical profile.

Thank you, changed in the manuscript.

L573: I don't see S/C attitude uncertainty included in the uncertainty equation. Every SC has attitude jitter. What is performance of Metop SC? Effect is smaller for larger SC, but it should be mentioned.

This is so far not considered in the uncertainty budget, since the attitude correction is not yet implemented for Metop in our system. After successful verification of the attitude correction, we will

address the accompanied uncertainty in more detail. We added the following sentence to the revised manuscript:

"Not yet taken into account is the spacecraft's attitude uncertainty, which will be addressed in more detail once the attitude correction in rOPS is included and verified. For larger spacecrafts like the Metop satellites this effect caused by attitude jitter is expected to be small, however."

L612: I don't understand this, 'carrier wave cycles in order of several cm.' Cycle slips can occur at ½ to N cycles, and therefore can be much larger than several cm.

The authors were referring to "an" undetected cycle slip, which is now corrected in the revised version of the manuscript:

"However, an undetected cycle slip can introduce a phase shift of half or full carrier wave cycles in order of a several centimeters, ..."

L617: I don't understand c = 0.001 m/min? So does this mean you apply an uncertainty due to cycle slips of 2 mm over a 2-min occultation event? This would be too small. Please explain.

We re-checked technical Metop documentation for a more reliable quantitative estimate. We did not come over a clear quantitative estimate but agree that the current setting was clearly too small. We now try to better reflect the effect by adopting a more plausible change of 1 mm/s, i.e., reflecting a 1% slip fraction per second relative to the half-cycle length (about 10 cm), more consistent with the documentation.

Therefore, to account for these undetected cycle slips as an estimated basic uncertainty, we include a change-rate factor $c$ = 1 mm/s, reflecting a 1% slip fraction per second relative to the half-cycle length. This leads to a gradual excess phase decrease (cumulative negative bias) with decreasing altitude from the time of highest altitude $t_{top}^{DLL}$ to lowest $t_{bot}^{DLL}$ in DLL measurement mode:

L619: use, Local spacecraft multipath.

We updated the manuscript accordingly.

This description of local sc mp is not very well substantiated. First of all, the period of local sc mp on orbit will be closer to the orbital period of 100 min. Also, the amplitude of the effect is related to the geometry of the antenna on the LEO sc and to surface property of the reflecting surface. We found with rough calculation that this effect can result in a phase error rate of close to 0.1 mm/s for a strong reflector. Since the geometry on the sc is fixed, these error may manifest as a basic systematic phase rate error that may not cancel in a climate study using an ensemble of occultations. This 'basic systematic phase rate error' aspect of local sc mp should be mentioned in the text.

Here, we model the effect of local spacecraft multipath during an occultation event (1-2 minutes duration), not to be confused with longer period orbital effects. We classify this effect as apparent

systematic uncertainty since with a changing viewing geometry from occultation to occultation the local multipath errors will average down when regional and temporal averages are calculated (Kursinski et al., 1997). The manuscript now includes previous studies using the corresponding model following GRAS specifications with a phase error rate of about 0.05 mm/s (Carrascosa-Sanz et al., 2003):

"The residual local multipath error effects on the phase measurements are modeled using a sinusoidal model, for representative broad beam antennas used in GNSS RO (Steiner and Kirchengast, 2005; Ramsauer and Kirchengast, 2001; Syndergaard, 1999). The sinusoidal shaped function is defined with a multipath phase error amplitude of 0.5 mm and period set to 60 seconds, resulting in multipath errors up to 1 mm, following GRAS-type error specifications (Carrascosa-Sanz et al., 2003). We classify this effect as apparent systematic uncertainty since with a changing viewing geometry from occultation to occultation the local spacecraft multipath errors will average down when regional and temporal averages are calculated (Kursinski et al., 1997)."

L622: Has this statement, 'can be reduced by modelling', been demonstrated experimentally for phase multipath on LEOs? 'The possible phase shifts of up to a few centimeters, introduced by local multipath, can be reduced by modeling the effect and the use of directional antennas'

The current statement was replaced and updated as follows:

"The possible phase shifts of up to a few centimeters, introduced by local spacecraft multipath, can be reduced by proper platform design and the use of directional antennas."

L627: The amplitude of 0.5mm used here seems incredibly small, and the period is much too small. The authors should provide a better justification for numbers or provide references.

For the Metop mission a dedicated effort was undertaken to diminish the effect of strong reflections and therefore a reduced amplitude is assumed. See also the comments above.

Figure 8: caption says thick orange line, but it is blue in the figure.

Fig8: Can you also include daily mean values (different color) in addition to daily median values? This will give some additional information on the impact of outliers.

Figure 8 now includes daily mean values (red) showing modest deviations compared to the daily median values (blue). The figure caption was corrected and updated accordingly.

L654: use, 'on average'

Thank you, changed in the manuscript.

L655: why do the Metop counts get smaller for the later missions?

The decrease in the number of profile counts are due to missing closed loop observations in the input data. We added the following declaration to the revised manuscript:

"Overall, on average, the daily number of profiles amounts to 647 profiles (Metop-A), 610 profiles (Metop-B), and 559 profiles (Metop-C). In the later 2020-JAS period EUMETSAT input data contains files with missing closed loop (either L2 or both frequencies), which reduces the number of processed event in this later study period."

L675-679 and Fig 10: It appears that the solar max period of 2013 has larger differences for all altitude regimes, as expected due to larger iono residuals. Is this statistically significant? This deserves some discussion from the authors in the text since ionospheric residuals are one of the largest challenges for RO especially at higher altitudes.

Although larger standard deviations can be observed in 2013 period, the box-whiskers in Figure 10 depicting the median (mid-line) and 16 % and 84 % percentile, the zero-line is still within +/- sigma and therefore not supporting the statistical significance. However, comparison against other solar max periods will be of interest in future, in order to learn more about the influence of ionospheric disturbances at excess phase level.

L686: Table 5, why do the orbit differences increase for the later 2020 period? Also, how do you go from the results in Table 5 to specifying the orbit velocity uncertainty above on L604, 0.02 mm/s? 0.02 mm/s seems too small for specifying the orbit velocity uncertainty.

The results in Table 5 represent inter-center differences of orbit solutions based on different software, parametrization, models, and input data, so including additional structural uncertainties, which are not represented in the orbit velocity uncertainty quantified in L604. We understand that the increase of the orbit differences in the later 2020 period is related to differences in the input data used. In the earlier 2013 period reprocessed GNSS orbit and clock data from CODE (repro2015; https://dx.doi.org/10.7892/boris.80011) of higher quality served as input for the precise orbit determination at WEGC, while in 2020 CODE final products (https://dx.doi.org/10.7892/boris.75876.4) were used. Additionally, in 2020-DJF the navigation tracking data from the GRAS zenith antenna were only available as hourly orbit dumps. The pre-processing and merge to a daily input file for the precise orbit determination somewhat diminish the quality further.

L692: Figure 11. How are the excess phase differences computed that are used to generate the stats in the figure? Are they the rms over the altitude region of interest? The maximum difference? Please state in text.

The statistics for each individual layer is calculated by averaging (mean) of the correspondent statistical measure used in the boxplot of Figure 11. We added the following text to L661:

"… layer averages (calculated from the vertical statistical measures, i.e., mean, median, stddev, percentiles, by obtaining the average over all values within the corresponding altitude layer):"

L763: How do the inter-center excess phase differences compare to your previously estimated excess phase errors? It looks like the inter-center differences are much larger than the earlier estimated uncertainties? Are these just due to mismatching (time/space) atmospheric differences? The estimates should be close to the inter-center differences. The authors should include more detailed discussion of the comparison between your estimated uncertainties and the inter-center differences.

In this study we aim for an uncertainty estimation of the observational data; the inter-comparison of excess phase data from different processing centers might include additional structural uncertainties due to the different processing schemes not captured in the uncertainty budget. In particular, the larger deviation of UCAR data in the lower troposphere indicates representation uncertainties from until yet unknown sources. The authors believe that the characteristics of the differences indicate an issue in the processing in the transition of closed loop and open loop measurements. To reassure this assumption. further research will be needed which could be a valuable future task within a broader inter-center comparison (similar to what, e.g., Steiner et al. (2020) did along the retrieval step from bending angle to atmospheric profiles retrieval).

We note that the results from the inter-comparison between excess phase data processed by EUMETSAT, WEGC, and UCAR experience larger differences in the lower troposphere than we quantified in the uncertainty budget. This indicates additional structural uncertainties arising from different processing schemes not captured in the estimated uncertainties of the observational data. In order to address this substantial differences a broader inter-center comparison study is advised.

L775: use, as a synthesis result.

Thank you, changed in the manuscript.

L782: Fig 16, Why is there no kink in the curve between SLTPA of -20 and -40 for the 2008 period, and there is a kink for the later periods? The authors should discuss this clear difference in the text.

We interpret this to originate in the different number of profiles as a function of altitude (Fig. 13), which possibly occurs due to changes in the on-board tracking and OL/CL transition handling of Metop within the different study periods. We added an explanation as follows:

"However, between an SLTP altitude of -20 km and -40 km, the two later periods exhibit a slight kink in the estimated random uncertainty compared to the 2008 period. This is presumably connected to the different structure of the corresponding numbers of profiles as a function of altitude shown in the lower panel of Figure 13 and changes in the receiver tracking."

---

## Author Comment (AC2)

**Response to Referee #2**

(***Referee report***: *https://doi.org/10.5194/amt-2023-28-RC2*)

**Manuscript:**

Innerkofler, J., Kirchengast, G., Schwärz, M., Marquardt, C., and Andres, Y.: *GNSS radio occultation excess phase processing for climate applications including uncertainty estimation*, Atmos. Meas. Tech. Discuss. [preprint], https://doi.org/10.5194/amt-2023-28, in review, 2023.
* * *
The authors thank the reviewer very much for the constructive and detailed feedback to the manuscript. We thoroughly considered all comments and carefully revised the manuscript accounting for them. Below are our point-by-point responses.

Comments by the reviewer are cited black upright, our responses are red. Line numbers used in our responses refer to the original AMT Discussions paper and text updates in the revised manuscript are quoted below in blue)

All citations referenced are provided in the bibliography of the revised manuscript.
* * *
**Summary**

This paper "GNSS radio occultation excess phase processing for climate applications including uncertainty estimation" by Innerkofler et al. describes a new RO excess phase processing system including excess phase uncertainty estimation for Metop series satellites. The excess phase profiles derived with such system are compared against those from different processing centers with different POD and excess phase algorithms. The uncertainty estimation of excess phase thus can be helpful to trace the excess phase errors back to SI standard. The main purpose of such "reprocessing" of operational missions is to provide climate quality data records. First there indeed exist the needs in RO community for reprocessing of Metop A/B/C excess phase datasets for inter-center comparison so that the structural uncertainties in the dry temperature can be traced back to excess phase or observational level. There are CDR products from EUMETSAT ROM SAF, but their RO products are based on existing excess phase profiles from other processing centers. UCAR CDAAC also has different versions of RO datasets for the same missions. Second, determining the uncertainty of RO bending angle retrievals are often limited to local spectral width (LSW), which is hard to connect with the excess phase uncertainty. Thus this study, the excess phase processing with uncertainty estimation, is scientifically important and a significant contribution to GNSS RO community, not only because it can be used for quality control for excess phase profiles, but also because the excess phase uncertainties can be further possibly quantified to derive the bending angle uncertainties. Technically, the processing system uses improved GNSS/Leo POD solutions, follows the standard excess phase processing procedure adopted by other missions/centers but with rigorous quality control. It uses the zero-differencing clock bias removal algorithm which depends on the ultra-stable clock onboard the Metop-

Satellites. The RO excess phase processing algorithm description is solid and covers all the related aspects. The quality control relies on the excess phase modeling, including geometric and atmospheric modeling, which is a significant step in excess phase algorithm. Overall, the excess phase processing of three three-months periods for Metop A/B/C shows successful reprocessing of Metop RO excess phase profiles, with uncertainty estimation, and within expected differences compared with datasets from other centers. The system can be used to generate a long time series for climate applications. Overall, this paper is well written and organized, with technical description in details, and presents the results clearly. The logic of the study is scientifically sound. The excess phase uncertainty estimation can be applied to other missions. I recommended this paper to be published at AMT after some minor revision.

Thank you for the valuable feedback, and your further specific suggestions below that helped to improve our manuscript.

One thing we want to point out here is that EUMETSAT ROM SAF, as application facility of the satellite data, has their own excess phase retrieval for the Metop satellites.

I am little bit concerned the large difference in excess phase in lower atmosphere (Figure 15) from UCAR. What's the main reason caused the large difference in standard deviation/biases? Is this related to how the excess phase model and/or the filtering/smoothing algorithm used for excess phase processing? The author should explain that in depth. Is this a proof that the actual uncertainties maybe larger than proposed (e.g: the STD compared with UCAR at 3.5 km MSL is more than 40cm, but the uncertainty in excess phase is less than 4cm.)?

In this study we aim for an uncertainty estimation of the observational data; the inter-comparison of excess phase data from different processing centers might include additional structural uncertainties due to the different processing schemes not captured in the uncertainty budget. In particular, the larger deviation of UCAR data in the lower troposphere indicates representation uncertainties from until yet unknown sources. The authors believe that the characteristics of the differences indicate an issue in the processing in the transition of closed loop and open loop measurements. To reassure this assumption. further research will be needed which could be a valuable future task within a broader inter-center comparison (similar to what, e.g., Steiner et al. (2020) did along the retrieval step from bending angle to atmospheric profiles retrieval). We included the following text in the summary & conclusion section of the manuscript:

"We note that the results from the inter-comparison between excess phase data processed by EUMETSAT, WEGC, and UCAR experience larger differences in the lower troposphere than we quantified in the uncertainty budget. This indicates additional structural uncertainties arising from different processing schemes not captured in the estimated uncertainties of the observational data. In order to address this substantial differences a broader inter-center comparison study is advised."

Though the excess phase uncertainty estimation is important, how this uncertainty can be translated into the Doppler shift and then into bending angle is not clearly mentioned. Some discussion on how

the excess phase uncertainty propagates further into bending angle should be given for the cases given. After all, the bending angle or derived temperature products are the Essential Climate Variable.

In the revised manuscript we now provide a paragraph discussing the propagation of the estimated uncertainties at excess phase through the subsequent retrieval chain. A detailed discussion of the uncertainty propagation from excess phase to bending angle is provided by Schwarz et al. (2018), which is one of the references in the manuscript. The added paragraph is as follows:

"The random and systematic uncertainty estimates at excess phase level are then propagated through the entire ODP retrieval chain in order to provide the final ECVs with their associated uncertainties. Additionally, the uncertainties quantified are employed in part of the retrieval operators of rOPS to improve the derivation of variables (e.g., ionosphere correction, statistical optimization, moist air retrieval). For details on the uncertainty propagation along this chain, starting from the estimates at excess phase level, see Schwarz et al. (2017, 2018); Schwarz (2018); Li et al. (2019)."

Technically, the excess phase processing in this study seem ignored both the GNSS and LEO satellite attitude information, please explain in detail how this can affect the error budget in excess phase.

Thank you for this relevant point; for the LEO attitude we considered nominal attitude with the assumption that for a satellite with a stable orientation like Metop the deviation compared to the application of measured or modeled attitude is small. However, within the ROM SAF validation activities it was found that this causes some residual biases at bending angle level (Alemany et al. 2022). However, with the scheduled implementation including quaternions in rOPS in a next update of the system this remaining weakness will be resolved. The impact of disregarding GNSS attitude is considered insignificant. Hunt et al. (2018) found that the effect of omitting GNSS attitude and phase center offsets amount for approximately 0.001 mm/s deviation in the Doppler shift, which is a negligibly small effect.

Figure 15, please explain why the STD profiles in figure 15 (e.g. left dotted line and right dotted line in any subfigure) are not symmetric even systematic bias approaches zero?

The STD shown in Figure 15 represents the 16 % and 84 % percentiles, indicating asymmetric distributions of the difference profile ensembles. We now note in the figure caption that percentiles are depicted.

**Minor Comments:**

Line 63, Is excess phase measurement accuracy/uncertainty really SI traceable given the excess phase model used, the GPS bit time series used, and the cycle slip correction uncertainty in the lower atmosphere?

In principle RO measurements are SI-traceable to the universal time standard. However, the use of auxiliary data in the processing and limitations in the tracking system, such as cycle slips, to some

degree impede SI-traceability and the quality of the measurements. With rigorous handling of the auxiliary data, the modeled excess phase, and uncertainty budget, we aim to keep these influences as limited as possible.

Figure 2, Are the attitude data belong to the auxiliary datasets from IERS? Aren't they provided by the mission operation?

Yes, the LEO attitude information is usually provided by the mission operator in form of either measured or simulated Euler angles or quaternions. In case of Metop, EUMETSAT provides a yaw-steering model (EUMETSAT2005) and UCAR corresponding daily attitude files. We updated Figure 2 accordingly.

Line 106, In table I, I believe the LEO attitude is important and should be labeled. It looks the usage of the LEO attitude is optional. But how could you convert the antenna offsets from space body frame to ECI without attitude/quaternion information?

As stated in line 394 of the manuscript, the attitude correction using quaternion information is not yet implemented for Metop in rOPS. The conversion of the antenna offsets is based on nominal attitude of the satellite with the assumption that for a satellite with a stable orientation like Metop this correction is small. However, within the ROM SAF validation activities it was found that this causes some residual biases at bending angle level (Alemany et al. 2022). The team is currently working on the implementation of the attitude correction in rOPS. Also, more detailed information is now included in the manuscript but we do not include LEO attitude information in Table 1, since it could not yet be used for the calculations in this study. The added information reads:

"Corrections for the changing orientation of the satellite in space and the deviation from nominal attitude during orbital revolution are not yet implemented in rOPS. However, although for missions with stable orientation like Metop this correction is small, it was found that not applying the correction introduces a small residual bias in bending angle data (Alemany et al., 2022). Therefore, it is treated as a priority to include this correction in a next version of rOPS. On the GNSS transmitter side, neither GNSS antenna offsets nor attitude are modeled, since they have a far smaller effect on RO processing than the LEO antenna offsets (Hunt et al., 2018)."

Line 130, does this reconstruction include the POD phase/pseudo-range also (RINEX files)?

No, the reconstruction does not include raw navigation tracking data from the RINEX files, but the receiver and transmitter clock biases as well as position and velocity determined in the precise orbit determination. These data are needed to reconstruct the raw occultation measurement time stamps and for un-differencing of the NCO phase from the EUMETSAT L1a data files.

Line 138, I believe UCAR have a more specific data address (url) to point to the exact location of the datasets used.

Right, thank you for the comment. The new CDAAC data interface contains DOIs of the respective datasets. This information has now been added to the manuscript. We also seized the opportunity to as well add more specific information on the EUMETSAT data. It now reads:

"In this study publicly available excess phase profiles from EUMETSAT (https://eoportal.eumetsat.int; 2008/2013-JAS: DOI: 10.15770/EUM_SEC_CLM_0015, processor: YAROS-1.4; 2020_DJF: processor: GRAS-4.6.2) and CDAAC (Metop-A: DOI: 10.5065/789w-m137, version: 2016.0120; Metop-B: DOI: 10.5065/1k0w-2272, version: 2016.0120; Metop-C: DOI: 10.5065/p8es-mc74; version: 2019.2580) have been used for such intercomparison (Sect. 4.2)."

Line 205, this has puzzled me. I think this is different from that used by UCAR. At the mean event time, the straight line may not be tangent to the WGS-84 ellipsoid surface. Unless the tangent point can be defined first and then the time difference (really small though) can be neglected. Please explain. How sensible the different profile location definition can affect the excess phase quality control (atmospheric modeling), especially at lower atmosphere?

Yes, we are using a slightly different definition for the occultation's reference point that is only depending on geometry and not on the atmospheric state or the specific duration of the actually tracked occultation event (and yes, this is why these mean tangent point locations can be computed beforehand and are independent of the occultation profiles retrieval process). Specifically, the mean event time is the time when the straight-line connection between GNSS and LEO is tangent to (i.e., just touches) the Earth's Ellipsoid, i.e., it is the time when the mean tangent point location is visited. We updated the description in the manuscript for better understanding as follows:

"The selected reference location of an event is defined on the Earth's ellipsoidal surface at the time when the straight-line connection between receiver and transmitter satellite is tangent to the Earth's surface (WGS-84/EGM2008, cf. Figure 3 for measurement geometry)."

Line 240, 'is used T and ln(p)', should be 'is used for T and ln(p)'? It would be better to give a reference for this interpolation scheme or a reason why these schemes are the best (e.g. ROPP manual compares different interpolation schemes). Different interpolation scheme can certainly affect how the bending angle bias look alike in different altitude.

Thank you, this is correct; the sentence was adapted in the manuscript. The horizontal 4-point polynomial (cubic) interpolation was adapted from a scheme originally implemented by M. E. Gorbunov (a detailed description was included in the PhD thesis of Lackner, 2010; Appendix A.3 therein). For vertical interpolation, several interpolation methods were compared and the most suitable selected. We included improved information on this as follows:

"Horizontal interpolation is performed by using a 4-point cubic-polynomial interpolation technique (for a detailed description see Lackner, 2010). For vertical interpolation of $T$ and $\ln(p)$ several interpolation

methods were compared (linear, cubic-spline, Savitzky-Golay filter) and the most robust fit through the nodes of the ECMWF altitude levels was selected. As a result, the vertical interpolation to the fixed-altitude grid $z$ is performed for $T$ and $\ln(p)$ using a natural cubic-spline interpolation, while the q profiles are interpolated linearly."

Line 260, the angle ζ should be between the velocity vector and the leo position vector, please label correctly in Figure. 3.

Thank you for noting, the angle ζ is now labeled correctly in the updated version of the manuscript.

Line 285, Carries phase can't be called phase pseudo-range, since the time measurements and phase measurements use different mechanisms in GNSS positioning techniques. It would be more appropriate to use its name 'carrier phase' than 'phase pseudo range.'

In line 285, the term "phase pseudorange" is just used in brackets, for introductory purposes, in order to also point to the terminology in the widely used book by Hofmann-Wellenhof et al. (2008). In line 287 we do clarify that more commonly the term "carrier phase" measurements is used. We don't use the term "pseudorange" in the remainder of the paper.

Line 337, Rewrite Eq. 5 to be consistent with Eq. 4, such as the ionosphere correction has opposite sign with atmospheric delay term, missed C in the third item in Eq. 5, inconsistent sign (+/-) between receiver/transmitter clock bias correction, etc.

Thank you for the detailed review of the equations. We corrected the formulae as proposed.

Line 346-349, some terms are misleading. You do not need to correct the clock bias, but to remove it. the antenna offset needs to be calculated in the proper coordinate system and added to the mass center of the satellites. The distance is not between satellites, but between receiver and transmitter antennas (pcvs/or offset since pcvs may not be used here). Please rewrite accurately.

Thank you, the text passage was re-written accordingly, as follows:

"This process includes the removal of receiver and transmitter clock biases, relativistic corrections, the calculation of the signal travel time, proper coordinate transformation of the antenna offsets and addition to the satellite's COM, and the calculation of the geometric distance between receiver and transmitter satellite antennas (see Figure 6)."

Figure 6 and from other context, why the LEO/GNSS satellites attitude input is optional? Are all the antennas offsets/pcvs defined in such a way that the attitude is not needed? How does this affect the error budget and excess phase itself? If attitude are not needed, you may also need to explicitly explain how the antenna offsets are applied.

Please also see the answer regarding the LEO attitude above.

Line 395-397, I do not understand the sentence ' common coordinate transformation from satellite body frame to ECI', isn't this the satellite attitude information (usually given as quaternions).

This is referring to the coordinate transformations of the LEO RO antenna offsets, which are defined in the satellite body frame assuming nominal attitude, to ECI. This step is needed to successfully and properly apply the offsets to the satellite's position and velocity (also given in ECI). As stated in Line 394-295, and discussed in other comments in this response to the reviewer, the attitude information is not yet taken into account, but will (in a next update of the system) be inserted exactly at this step of the processing.

Line 410: Here the time delay correction does not consider the GNSS antenna offset. There is neglected time bias of about 2/C (assume GNSS antenna offset length of 2m) about 7 ns. Please justify how this affects the excess phase calculation with zero-differencing methods. For single differencing, this may be absorbed by differencing itself.

This is considered to be a small effect, since the pointing of the GNSS antenna is very stable over the short duration of an occultation, and the non-time dependent part of this effect will cancel out in the derivation of the excess phase to Doppler. Hunt et al. (2018) found that the effect of omitting GNSS attitude and antenna offsets only amount to approximately 0.001 mm/s deviation in the Doppler shift, which is a negligibly small effect.

Line 436-437, down sampling of RS data to 50HZ, the authors used the 20 samples arithmetic mean. Please explain how the 20 samples arithmetic mean affect the cycle slips (if not corrected yet) especially for lower atmosphere.

The cycle slip correction is performed right before the down-sampling based on the 1000 Hz RS (I/Q) data. Cycle slips remaining undetected by this step will increase the arithmetic mean of the respective sample and can only be accounted for in the uncertainty budget.

Line 453, the Metop POD antenna are not designed to track high rate GNSS signals thus the single differencing may not be effectively used (as COSMIC does) with low rate POD antenna observations.

Exactly, for the application of the single differencing method for the Metop mission, the reference link data measured at the zenith antenna with 1 Hz would need to be up-sampled to match the 50 Hz high-rate occultation link data. This introduces additional noise and reduces the quality of the calculated excess phase data. Therefore, for RO missions with ultra-stable onboard oscillators (USOs, like on Metop), the preferred method is to eliminate the LEO clock errors using the clock biases estimated in POD (i.e., to apply zero differencing).

Line 505, Aren't the sampling rate defined at the receiver time with a constant interval? Please explain what caused the drift.

The raw measurement time stamps of GRAS exhibit a frequency offset and irregularities due to external temperature changes (Montenbruck et al., 2008). The frequency offset is corrected for and resulting time stamps are (very) slightly deviating from nominal sampling rate. From Table 4 we can see that for Metop data this obviously is not an issue, however, since for no Metop profile considered in this study the sampling check failed (i.e., we observe a rejection rate of 0 %).

Line 549, what's the criteria to use 7.5 m/s.

The value was empirically derived by detailed sensitivity examinations from analyses of event ensemble from multiple RO missions (done within the work of Seidl 2018).

Line 614, DLL already defined at line 310.

Thank you, now the acronym is introduced only at line 310.

Line 619, please define the $t_{bot}^{DLL}$ and $t_{top}^{DLL}$. This looks quite small. Given one minute of travel time in the lower atmosphere, the cycle slip error is only 0.001m=1mm? If this is true, how do you explain large excess phase difference in lower troposphere between different processing centers?

Regarding the larger excess phase differences in the lower troposphere, please see the "main comments" section of this review above. However, we re-checked technical Metop documentation for a more reliable quantitative estimate. We did not come over a clear quantitative estimate but agree that the current setting was clearly too small. We now try to better reflect the effect by adopting a more plausible change of 1 mm/s, i.e., reflecting a 1% slip fraction per second relative to the half-cycle length (about 10 cm), more consistent with the documentation.

Therefore, to account for these undetected cycle slips as an estimated basic uncertainty, we include a change-rate factor $c$ = 1 mm/s, reflecting a 1% slip fraction per second relative to the half-cycle length. This leads to a gradual excess phase decrease (cumulative negative bias) with decreasing altitude from the time of highest altitude $t_{top}^{DLL}$ to lowest $t_{bot}^{DLL}$ in DLL measurement mode:

Line 655, Why does the Metop-C have less daily RO profiles than Metop-A/B?

In the 2020-DJF period Metop-B/C feature a similar number RO profiles. In the earlier measurement periods 2008-JAS and 2013-JAS we can observe larger numbers of RO profiles per satellite.  The decrease in the number of profile counts in the later 2020-JAS period is due to missing closed loop observations in the input data. We added the following declaration to the revised manuscript:

"Overall, on average, the daily number of profiles amounts to 647 profiles (Metop-A), 610 profiles (Metop-B), and 559 profiles (Metop-C). In the later 2020-JAS period EUMETSAT input data contains

files with missing closed loop (either L2 or both frequencies), which reduces the number of processed event in this later study period."

Line 731-732, it is hard to follow what the authors talk about. Are they trying to compare the setting and rising differences in total profile numbers or to compare inter-center difference?

Here we refer to Figure 13 and the total number of difference profiles depicted as function of altitude therein. As described it can be observed that differences between the number of setting and rising occultations exist in dependence of the altitude.

---

## Author Comment (AC3)

**Response to Referee #3**

*(**Referee report**: https://doi.org/10.5194/amt-2023-28-RC3)*

**Manuscript:**

Innerkofler, J., Kirchengast, G., Schwärz, M., Marquardt, C., and Andres, Y.: *GNSS radio occultation excess phase processing for climate applications including uncertainty estimation*, Atmos. Meas. Tech. Discuss. [preprint], https://doi.org/10.5194/amt-2023-28, in review, 2023.
* * *
The authors thank the reviewer very much for his comments, which helped to further improve the manuscript, and his commendation. We thoroughly considered all comments and carefully revised the manuscript accounting for them. Below are our point-by-point responses.

Comments by the reviewer are cited black upright, our responses are red. Line numbers used in our responses refer to the original AMT Discussions paper and text updates in the revised manuscript are quoted below in blue)
* * *
**Summary**

This manuscript provides an overview of Level 1a (L1a) processing of radio occultation (RO), a thorough description of the authors' newly-developed processing system, and demonstration of the quality of said processing system. L1a processing transforms the raw measurements between GNSS satellites and low-earth orbiting RO receivers into measurements of excess phase –the phase "added" by propagation through the atmosphere. Precise estimation of this excess phase is critical to the processing chain of occultation events from GNSS signals to, e.g., temperature –a so-called essential climate variable (ECV). Thorough documentation, quality control, and uncertainty estimation is important for the community to understand what was done and why, and how the choices in processing affect the resultant ECVs.

I acknowledge and applaud the authors for this manuscript. The advancement of RO science at their center is evident in their completeness of the document. I certainly learned a great deal about the steps that go into excess phase retrieval, and I believe that is an important contribution to our subfield and, more broadly, atmospheric science. I think the quality of the retrieval is showcased nicely in their analysis in section 4. And, should they be coming, I look forward to the next publication(s) on later steps in the RO processing chain.

My primary, minor comment for the authors to address relates to the use of ERA5 analysis for their sensitivity analysis. Typically, one thinks that comparing an observation to an analysis that assimilates that observation would lead to artificially small difference statistics. I don't suspect that the use of, say, ERA5 forecasts instead will lead to significant changes in the calculated O-B values. But, I do think it warrants consideration and some discussion.

Regarding the use of ERA5 analysis used in the sensitivity analysis the authors are aware that the analysis assimilates RO measurements as stated in Line 150 of the manuscript. However, the retrieval uses the independent ERA5 short-range forecast fields for the extraction of the modeled excess phase profiles used in the quality and uncertainty evaluation. We also do recognize that the use of ERA5 short-range forecasts instead of the analysis will not change the calculated O-B values significantly and regard the use of the ERA5 analysis for the sensitivity analysis sufficient. However, we of course do plan to incorporate independent datasets in future dedicated validation studies.

**Line-by-line comments**

Line 35: suggest "By observing Essential…"

Line 56: suggest "Evaluation of basic…"

Line 78: suggest "subdivided into sensitivity…analyses"

Line 152: suggest "…data suit the…"

Line 154: suggest removing "in order"

Line 175: "that builds on"

Thank you, we updated the manuscript following the suggestions above.

Line 205: suggest "surface at the point…line connects between"

The description of the mean tangent point location in space and time was rewritten based on a comment by another reviewer, in order to better describe the actual calculation of this occultation event reference location:

"The selected reference location of an event is defined on the Earth's ellipsoidal surface at the time when the straight-line connection between receiver and transmitter satellite is tangent to the Earth's surface (WGS-84/EGM2008, cf. Figure 3 for measurement geometry)."

Line 207: apologies if I missed it, but is the mean tangent point identified in Fig. 3? It's not needed to add it to the figure if it is not already there.

The term tangent point might be misleading, since it reflects a reference point at the time when the straight line connection between GNSS and LEO is tangent to (i.e., just touches) the Earth's Ellipsoid. Figure 3 depicts the schematic of the occultation event geometry at that point. We added the following description to the figure caption and identify the mean tangent point in the figure:

"The schematic depicts the occultation event at the time when the straight-line connection between GNSS and LEO satellite is tangent to the Earth's ellipsoidal and defines the mean tangent point (MTP) in this way."

Line 222: "In principle"

Line 278: recommend removing "also"

Line 279: recommend removing "level"

Line 304: "occultations that may"

Line 320: closed parentheses that is unmatched

Thank you, we updated the manuscript following the recommendations above.

Line 357: the sentence ends with a fragment. Consider rewording.

Rewrote, the sentence now reads as follows:

"The atomic clocks aboard the GNSS satellites are considered stable over the short duration of an occultation event of approximately 1 to 2 minutes with an accuracy of between $10^{-11}$ and $10^{-12}$, while more up-to-date clocks feature even higher accuracy (Griggs et al., 2015; Hauschild et al., 2013)."

Line 442: given that Eq. (9) has a factor 2k*pi, should this be "are always within 2pi"?

This was corrected in the manuscript and now states:

"… are always within $\pm\pi$: …"

Fig. 7: the middle and right panels cut off quite a bit of interesting information due to the bounds at +-15 cm. Is it possible to expand/shift these bounds while retaining resolution of some of the fine-scale features from 10+ km?

I am afraid not, since we already tried to find the best match between indicating the characteristics below an altitude of 10 km and not missing the fine-scale variations at higher altitudes. We hence preferred to keep this as is.

Line 493: "Schwarz et al." ends up in parentheses when it should be outside of them.

Thank you, changed in manuscript.

Table 4: if it is sensible to do, please include the total rejection fraction in the table caption.

The total rejection rate is now included in the updated version of the manuscript:

"Overview on the rOPS L1a quality processing. All parameters apply to Metop/GRAS data. Middle column separates the total data rejection rate $f_{QC}$ of 4.17 % for all data (9 months) processed by WEGC for this study in rejection fractions for every single quality control step."

Line 540: suggest "profile is done by checking"

Line 567: "despite averaging"

Line 656: suggest "worthwhile to take into account"

Line 664: references are given in Bibtex format

Line 665: suggest removing "core-strength" or rewording

Line 682: references inside parentheses have parentheses

Thank you, we updated the manuscript following the comments above.

Section 4.1.3: what are the counts of the co-located Metop profiles used in this section? The robustness of the results is not clear without that information. If it was provided elsewhere, please reference to it.

The information was now added in the latest version of the manuscript:

"This results in an adequate number of co-located profiles for robust statistics: Metop-B vs. Metop-A: 2708 (2013-JAS), 1641 (2020-DJF); Metop-C vs. Metop-A: 1797 (2020-DJF)."

Lines 717-718: "with the help of which the RO events" is not clear. Consider rewording.

The authors rewrote this part in order to make it more clear:

"Prior to the comparison and the calculation of profile differences, the external L1a profiles from EUMETSAT and UCAR were converted to rOPS-L1a format, including the calculation of consistent and unambiguous event identifier information and some format alignments of the datasets (Section 2.2). This enables to reliably match the RO events from the three different processing centers."

Fig. 11 and text: is this given as WEGC minus EUMETSAT? Please make this clear.

The figure header states "WEGC-Bernese vs. EUMETSAT" and we updated the figure caption in order to provide a better description:

"Sensitivity analysis results from comparing WEGC rOPS L1a excess phase profiles from Metop-A (blue), Metop-B (red), and Metop-C (black) based on WEGC-Bernese orbits minus excess phase profiles based on EUMETSAT POD solutions, …"

Also the text in Lines 694-695 was revised:

"Comparison results of rOPS-processed excess phases based on WEGC-Bernese orbit processing against using EUMETSAT POD exhibit a similar agreement across all vertical layers, time periods and satellites."

Line 779: is there an example percent or range of representative percents that could be given?

A representative percentage envelope number was included in the updated text of the manuscript:

"In the mesosphere and upper stratosphere, the estimated random and systematic uncertainty components are small at millimetric level, while towards lower altitudes, where the absolute excess phase increases rapidly, the estimated random uncertainty component amounts to several centimeters (which is still very small at levels below 0.02 % in relative terms in the troposphere, though)."

---

## Author Response (AR2)

**Author's Response**

**Manuscript:**
Innerkofler, J., Kirchengast, G., Schwärz, M., Marquardt, C., and Andres, Y.: *GNSS radio occultation excess phase processing for climate applications including uncertainty estimation*, Atmos. Meas. Tech. Discuss. [preprint], https://doi.org/10.5194/amt-2023-28, in review, 2023.
* * *
We revised the manuscript along the editors comments and addressed the final issues. Additionally, we replaced "in submission" or "in press" citations with the final publication references. The authors thank the editor for the professional handling of the manuscript.
* * *
**Comments by the editor**

The authors have done a hard work in answering the comments made by the reviewers and modifying the manuscript. However, I see that two important requests have not been totally tackled. Before final acceptance they must be addressed:

1) Referees 1 and 3 are concerned about the use of ERA5 for validation of RO data and request consideration and discussion of this issue. In the new version of the manuscript a sentence has just been extended on this subject. More explanations are needed.

Ok, we further extended the discussion in Section 2.1 on this subject as follows:

"Although the assimilation of RO data has an influence on the ERA5 analysis data, we consider this analysis suitable for supporting the task of evaluation of the quality and robustness of the implemented excess phase processing in this study. We note that for actual validation of data, which is not the focus here, further independent datasets should be used."

In addition, we checked the text of the whole manuscript whether we inadvertently had used the term "validation" instead of "evaluation" at other places. We found three cases and either replaced the term by "evaluation" or found we can drop it in the context of these sentences.

2) Referee 1 states: "Another concern is the 3 data periods used in the study – they do not cover the most challenging regimes that RO has to track in. Analyzing a more challenging period may shed more light on RO error sources and uncertainty and better inform the community.". Later he also writes: "If the authors can't include analysis of these data, then they should at least discuss any issues seen in these more challenging data.". The authors replied: "After careful consideration however, we came to the conclusion that the test data periods chosen serve the demonstrative purpose of this introductory paper of the L1a excess phase processing including uncertainty estimation and that reprocessing would be beyond the scope of this purpose." So, what have been the careful considerations of the authors and their conclusions ? And in addition, why would they preclude the authors from even taking into account the request by the reviewer to "at least discuss any issues seen in these more challenging data".

We chose data periods for our study that are representative for different solar cycle conditions, with July to September 2013 representing a period of maximum solar activity with increased challenges for RO signal tracking. In comparison to the other periods of low solar activity, the evaluation of the sensitivity analysis, the rejection rates of the quality control, and estimated uncertainties all showed rather similar behavior. This indicates that the presented algorithmic setup is robustly working over a broad range of geophysical conditions, mainly targeting the rejection of "really un-physical" excess phase profiles for a climate-oriented data processing.

However, as a further crosscheck, we now performed basic analyses of two additional months, October 2013 and January 2014, in the challenging equatorial plasma bubble season. The analysis revealed no major differences again, compared to the data investigated in the manuscript, and also differences in the uncertainty budget for those periods are found to be small. For a truly detailed analysis of different ionospheric conditions, a dedicated study could provide more insights (which we consider beyond the scope of this study, which is already rather comprehensive).

To include information on our further crosscheck, we added the following sentence (*italic*) to the appropriate paragraph in the manuscript:

"In this study we used three multi-month time periods as basis for the assessment, each comprising 3 months: July to September 2008 (Metop-A); July to September 2013 (Metop-A/B); December 2019 to February 2020 (Metop-A/B/C). These are representative for different solar cycle and summer/winter conditions over more than a decade from 2008 to 2020. *As a cross-check, we also investigated data in the equatorial plasma bubble season (September 2013 to March 2014), which involves particularly challenging geophysical conditions, but found no appreciable differences in comparison to the three time periods chosen.*"

---

## Author Response (AR3)

**Author's Response**

**Manuscript:**

Innerkofler, J., Kirchengast, G., Schwärz, M., Marquardt, C., and Andres, Y.: *GNSS radio occultation excess phase processing for climate applications including uncertainty estimation*, Atmos. Meas. Tech. Discuss. [preprint], https://doi.org/10.5194/amt-2023-28, in review, 2023.
* * *
As advised, the authors rechecked all figures of the manuscript for interpretation for readers with color vision deficiencies and updated Figures 4, 5, 7, 8, and 9 accordingly.